# Three types of remapping with linear decoders: A population-geometric perspective

**Guillermo Martín-Sánchez, Christian K. Machens** [ID] ⌾, **William F. Podlaski** [ID] ⌾*

Champalimaud Neuroscience Programme, Champalimaud Foundation, Lisbon, Portugal

⌾ These authors contributed equally to this work.
* william.podlaski@research.fchampalimaud.org

**Data availability statement:** All relevant data are within the manuscript and its Supporting Information files. Code is available on GitHub at

## Abstract

Hippocampal remapping, in which place cells form distinct activity maps across different environments, is a well-established phenomenon with a range of theoretical interpretations. Some theories propose that remapping helps to minimize interference between competing spatial memories, whereas others link it to shifts in an underlying latent state representation. However, how these interpretations of remapping relate to one another, and what types of activity changes they are compatible with, remains unclear. To unify and elucidate the mechanisms behind remapping, we here adopt a neural coding and population geometry perspective. Assuming that hippocampal population activity can be understood through a linearly-decodable latent space, we show that there are three possible mechanisms to induce remapping: (i) a true change in the mapping between neural and latent space, (ii) modulation of activity due to non-spatial mixed selectivity of place cells, or (iii) neural variability in the null space of the latent space that reflects a redundant code. We simulate and visualize examples of these remapping types in a network model, and relate the resultant remapping behavior to various models and experimental findings in the literature. Overall, our work serves as a unifying framework with which to visualize, understand, and compare the wide array of theories and experimental observations about remapping, and may serve as a testbed for understanding neural response variability under various experimental conditions.

## Author summary

Place cells of the hippocampus form unique activity patterns in different environments, a process called remapping. However, it is not clear what the relationship is between changes in place cell activity and the underlying signals that the hippocampus represents. This study presents a new framework using population geometry and neural coding principles to explain hippocampal representations, and identifies three possible causes of remapping: true changes in how variables are represented, responses to non-spatial

the following link: https://github.com/guillemarsan/RemappingGeometry.

**Funding:** This work was supported by the Simons Collaboration on the Global Brain (543009 and 2794-04) and NIH R01 EY035896 and NIH RF1 NS127107 to CKM. The funders had no role in study design, data collection and analysis, decision to publish, or preparation of the manuscript.

factors, or non-coding neural noise. Simulations and visualizations illustrate these mechanisms and connect them to various experimental and theoretical results, providing a tool to better understand memory, navigation, and neural variability.

## Introduction

Place cells in areas CA1 and CA3 of the hippocampus exhibit localized, spatial firing patterns, which are thought to contribute to a cognitive map of space [1,2]. However, hippocampal activity is known to be modulated by many other variables besides the animal's position [3,4]—this includes other types of spatial information (e.g., head direction), sensory information, task-related information, internal states, and changes in context. Perhaps the most striking manifestation of this modulation is remapping—the phenomenon in which place cell activity appears to form distinct representations, even in response to relatively minor environmental changes [5,6]. Remapping comes in different flavors—it can be 'complete', in which the activity appears to change randomly and globally across an entire environment [7,8], or it may be 'partial', in which some neurons change their spatial preference, while others only exhibit minor changes in firing rate (i.e., rate remapping; [9–12]). The implications of remapping on the hippocampal code are still unclear.

Historically, explanations for remapping have focused on the perspectives of spatial navigation and memory [4,5,13,14]. The definitive model of such a perspective is that of the "multi-chart" continuous attractor network [15,16], in which the hippocampus is hypothesized to store a set of spatial maps, or "charts", as attractors of the network dynamics. The theory dictates that each chart should be random and uncorrelated with all others in order to promote high memory capacity and reduced interference between different maps [17]. The multi-chart model thus predicts random global remapping, consistent with several experimental studies observing orthogonalized representations in CA3 (e.g., [18–20]). However, it is less clear how this spatial memory perspective can account for partial or rate remapping, mixed-selective and purely non-spatially selective cells (e.g., [21–26]), as well as place cells with multiple fields [27,28].

An alternative perspective takes the spatial focus of the original cognitive map theory and expands it to a more generic internal representation of the environment [29]. Recent theories based on this approach suggest that the hippocampus builds a latent state space of the environment in order to solve tasks [30,31]. While this latent space view predicts similar spatial tuning profiles as the spatial memory view, it has the benefit of providing a more explicit functional role for place cell mixed selectivity, and is also compatible with non-spatially-selective cells and multiple fields. Some theories view place cells as representing a conjunction of space and other (e.g., sensory) variables [32], while other theories view space as implicitly constructed through a latent sequence of hidden states [33–38]. The latent space view can also account for different types of remapping [39], and may lead to more structured, non-random remapping effects as compared to the predictions of the spatial memory perspective [32]. However, such remapping questions have not yet been systematically quantified from this perspective, and questions remain about the specificity of its predictions and their compatibility with the spatial memory view.

While place fields and remapping have traditionally been understood at the single-neuron level, the recent shift in focus towards a population-level view of neural coding [40–42] may help to clarify and unify these varying perspectives on hippocampal activity and function [43]. The population view has been applied to hippocampal representations (e.g., [44–46]), but its implications for remapping are not yet clear (but see [6,47]; see Discussion). In this

work, we use a neural coding and population geometry perspective to develop a theoretical framework for hippocampal remapping. Under the assumption of a linearly-decodable latent space, we show that there are three fundamental mechanisms that can explain remapping, which we term encoder-decoder, mixed-selective, and null-space remapping. We explain how various experimental and theoretical findings partition into them. Rather than ruling one mechanism out in favor of another, we suggest that all three remapping mechanisms are likely to be accurate depictions of hippocampal activity changes under different conditions, and propose our framework as a useful perspective with which to understand the variability of hippocampal representations within and across environments.

## Results

We start from the standard setup of spatial navigation and remapping, in which an animal navigates through several environments while hippocampal place cell activity is monitored. This setup is schematized for two linear tracks *A* and *B* in Fig 1a and 1b. We consider that these environments are represented by the animal through a common set of environmental variables. These variables include (1-d or 2-d) spatial position, $\mathbf{p}$, and potentially one or more additional cognitive variables, $\mathbf{c}$, such as internal (e.g., behavioral) or external (e.g, sensory) states. For simplicity, we assume that these cognitive variables have a deterministic relationship to position for a given environment, which we denote as $\mathbf{c}(\mathbf{p})$. Our aim is to model how the environmental variables are *encoded* into hippocampal population activity across environments, and, analogously, how estimates of these variables may be *decoded* from such activity (Fig 1a). We will present this aim in three steps. First, we will define how firing rates vary with position in each environment, via a firing-rate map, $\mathbf{r}(\mathbf{p})$, that links the environmental variables to neural activity through an intermediate latent space. Second, we will specify a particular mechanistic autoencoder network architecture that is compatible with these activity maps and latent space representations. Third, we will examine how the constraints imposed by the latent space affect activity changes across environments (i.e., $\mathbf{r}^A(\mathbf{p})$ vs. $\mathbf{r}^B(\mathbf{p})$ in Fig 1b, right).

**Mapping spatial position to neural activity through a latent state space.** Many encoder-decoder mappings could relate environmental variables such as spatial position to neural activity (Fig 1a; [48–51]; see Discussion). The central feature of our model is the introduction of a low-dimensional latent space, $\mathbf{z}(\mathbf{p})$, as an intermediate representation between the environmental variables and neural activity (Fig 1c). This latent space can be interpreted as an internal hippocampal representation of the environmental variables. To build a firing-rate map for a given environment, $\mathbf{r}(\mathbf{p})$, we therefore first define a mapping from the environmental variables to the latent space, $(\mathbf{p}, \mathbf{c}(\mathbf{p})) \leftrightarrow \mathbf{z}(\mathbf{p})$ and then a mapping from the latent space to the neural activities, $\mathbf{z}(\mathbf{p}) \leftrightarrow \mathbf{r}(\mathbf{p})$.

First, the mapping from environmental variables to the latent space accounts for the fact that the internal representation of spatial position and other quantities need not reflect Euclidean coordinates, but can be deformed or nonlinear. In fact, many previous constructive models of place and grid cells consider position as a curved space parameterized by one or more angles ([52]; see Discussion), which we adopt here (Methods 1.3). Concretely, for a linear track, spatial position is mapped to an angle ($\alpha_p(p)$, Fig 1d), forming a circular trajectory in 2-d latent space ($\mathbf{z_p}(p)$, Fig 1e). Adding a cognitive variable $c(p)$ (e.g., odor concentration), extends the angle representation to (($\alpha_p(p), \alpha_c(p)$), Fig 1g), which then maps onto a four-dimensional latent space ($\mathbf{z_p}(p), \mathbf{z_c}(p)$), with trajectories confined to a torus (shown as a 3-d nonlinear embedding of the 4-d space in Fig 1h). This picture then generalizes to higher dimensions, with $P + C$ environmental variables mapping onto hypertoroidal trajectories in $Z = (2P + 2C)$-dimensional latent space. As we will see below, an angular code is a simple and

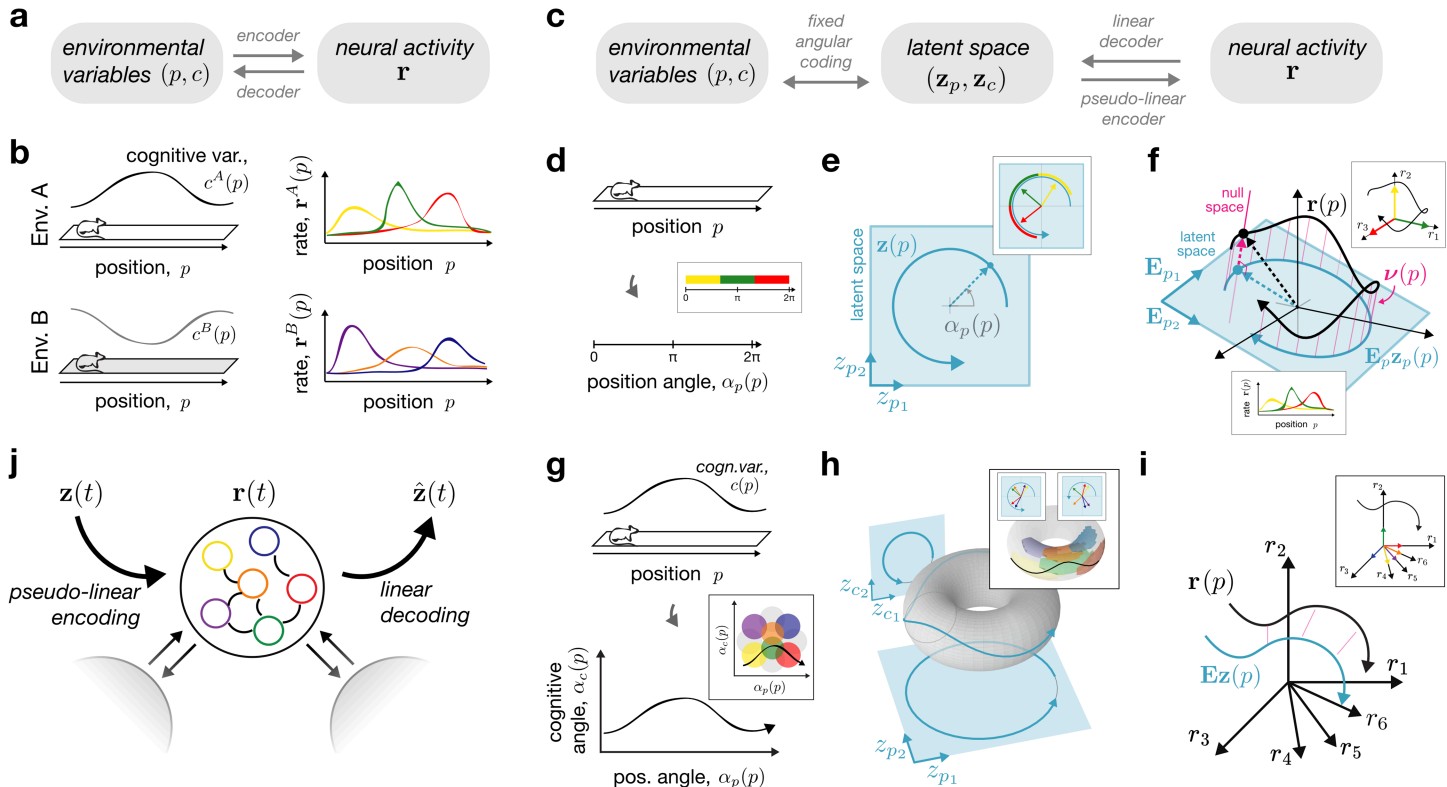

**Fig 1. A neural coding and population-geometric perspective on hippocampal place fields. a,b**: General overview of spatial coding and remapping; two linear track environments, *A* and *B*, are characterized by a common set of environmental variables (position, *p*, and cognitive variable(s), *c*, which may be environment-specific, $c^A(p)$ and $c^B(p)$); spatial and cognitive variables are related to place cell firing-rate maps ($\mathbf{r}^A(p)$ and $\mathbf{r}^B(p)$) through encoder and decoder mappings. **c**: The constrained model considered in this work, featuring an intermediate latent space, $\mathbf{z} = (\mathbf{z_p}, \mathbf{z_c})$, with two assumptions: (1) fixed angular coding from environmental variables to latent space, and (2) environment-specific linear decoding from neural activity to latent space. **d**: 1-d position *p* is encoded as the position angle $\alpha_p(p)$. **e**: This gives rise to circular trajectories in a 2-d latent space $(z_{p_1}(p), z_{p_2}(p))$. **f**: The activity trajectory in neural state space ($\mathbf{r}(p)$, black) can be seen as the combination of a linear encoding of latent space ($\mathbf{E_p z_p}(p)$, blue), plus a null space component ($\boldsymbol{\nu}(p)$, pink). **d,e,f insets**: Sequential place field activity (**f**, bottom inset) can be seen as the alignment of the trajectory with each neuron's axis (**f**, top inset); the linear decoder results in a rotation of each neural axis in latent space (**e**); each area of the latent trajectory (**e**) or angle space (**d**) is colored by the most active neuron (Methods 3.5). **g,h,i**: Same as panels (**d,e,f**) but for a pair of position and cognitive variables (*p,c*), leading to the pair of angles $(\alpha_p(p), \alpha_c(p))$, a four-dimensional latent space with trajectories confined to a 4-d torus (shown as a 3-d embedding), and the same neural trajectory in rate space (black) composed of the combination of latent space (blue) and null space (pink). **g,h,i insets**: As above, neural tuning can be visualized in latent and angle space, this time resulting in localized surface patches for each neuron on the torus or in angle space (Methods 3.5). **j**: Place-field representations are modeled in an autoencoder recurrent neural network (RNN) following Eq 3 and compatible with Eqs 1 and 2 (Methods 3.2).

sufficient model for generating localized place-like firing fields. However, we stress that it is not necessary for our theoretical results, which are compatible with any one-to-one mapping between environmental and latent variables (Methods 1.3). Importantly, because this mapping specifies the "coordinate system" of the internal representation, we consider it to be fixed across all environments.

Next, we impose a constraint on the mapping from latent space to neural space. The constraint stems from an assumption about the reverse mapping, from neurons to latents. Specifically, we assume that the latent variables can be *linearly decoded* from neural activity as

$$\mathbf{z}(\mathbf{p}) = \mathbf{Dr}(\mathbf{p}), \tag{1}$$

where $\mathbf{D}$ is a $Z \times N$ decoding matrix that maps the $N$ neural activities into the $Z$ latent variables (Methods 1.1). This choice is analogous to a linear population vector code [49], and can be viewed as projecting the full $N$-dimensional neural trajectory onto a linear subspace. Moreover, a weighted linear sum is also a plausible model for what any downstream neuron can read out from the network [53,54]. Given more neurons than latent variables, $N>Z$, the linear readout has many possible inverses, i.e., there are now many possible linear or nonlinear encodings from the latent space to the neural activities. However, we can specify the most general encoding model consistent with Eq 1, which we call the *pseudo-linear encoder* (Methods 1.1), and write it as

$$\mathbf{r}(\mathbf{p}) = \mathbf{E}\mathbf{z}(\mathbf{p}) + \boldsymbol{\nu}(\mathbf{z}(\mathbf{p}); \mathbf{E}), \tag{2}$$

where $\mathbf{E} = \mathbf{D}^+$ is the right pseudo-inverse of the decoding matrix from Eq 1, and $\boldsymbol{\nu}(\cdot)$ is an arbitrary nonlinear function in the null space of $\mathbf{D}$, that depends on the particular mechanistic network model. Accordingly, the first term on the right-hand-side of Eq 2 constrains $Z$ dimensions of $\mathbf{r}$ to be linearly related to $\mathbf{z}$, and the second term on the right-hand-side specifies the other $N-Z$ dimensions and accounts for any nonlinearities in the encoding (e.g., non-negativity; [54,55]). A geometrical interpretation of the pseudo-linear encoder for a linear track is given in Fig 1f. Here, the neural trajectory, $\mathbf{r}(p)$, is composed of the latent trajectory embedded in neural state space (Fig 1f, blue), plus the addition of an orthogonal, null space component (Fig 1f, pink). In turn, the linear decoder effectively collapses the full neural trajectory (black curve) onto the 2-d latent plane (blue curve), thereby removing variability in the third, null space direction (pink). The same geometric intuition extends to higher-dimensional settings (Fig 1i).

An advantage of linear decoding and pseudo-linear encoding is that it provides a straightforward way of visualizing neural firing fields in the space of environmental variables. To see this, we can return to Fig 1d–1f, and observe that the sequential place-field activity (Fig 1f, bottom inset) can be explained by the temporary alignment of the neural trajectory with each neuron's axis (Fig 1f, top inset, colored arrows). The linear decoder then projects these axes onto the latent space (Fig 1e, inset, colored arrows). By coloring segments of the latent or angle-space trajectories with the most active neuron (Fig 1d and 1e, insets), we generate a visual map of tuning preferences overlaid on the environmental and latent trajectories (Methods 3.5). From this view, it becomes clearer why an angular encoding is so suitable for generating place fields, as the curved latent trajectory temporarily aligns with particular neurons' preferred tuning vectors in localized areas. Similar visualizations can be made for higher-dimensional representations (Fig 1g–1i, insets), resulting in a tessellated pattern of tuning preferences on the toroidal manifold in latent space (Fig 1h, inset, colored circles), and in angle space (Fig 1g, inset, colored circles; see [46,56] for similar visualizations).

**Nonlinear encoding and linear decoding in an autoencoder network.** The encoding/decoding perspective naturally suggests modeling place fields with an autoencoder network (Methods 1.1). Such a network nonlinearly encodes a latent variable input, $\mathbf{z}$, and then enables an estimate, $\hat{\mathbf{z}}$, to be linearly decoded from the activity (Fig 1j). We utilize an established recurrent neural network (RNN) model (Methods 3.2; [57–59]), which comes with additional, biologically-plausible characteristics (see Discussion). Importantly, this model satisfies the encoding and decoding constraints above in Eqs 1 and 2, and its weights are set to optimally represent a chosen set of latent variables via the constrained optimization problem

$$\mathbf{r}^*(\mathbf{p}) = \arg\min_{\mathbf{r}} ||\mathbf{z}(\mathbf{p}) - \mathbf{Dr}||^2 + 2\mathbf{T}^\top\mathbf{r}$$

$$\text{subj. to} \quad \mathbf{r} \geq \mathbf{0},$$

(3)

with the second term in the objective acting as a regularizer, and $\mathbf{T}$ being a vector of thresholds or biases for each neuron. This network-level encoding model can be seen as a more explicit version of Eq 2, which specifies the nonlinear null space term ($\boldsymbol{\nu}$) by constraining the neural firing rates to be non-negative with a $\ell_1$-cost on activity (Methods 3.2). In practice, we simulate the RNN's activity in response to latent space trajectories as input, and then decode estimates of the latent trajectories in the output. Random tuning in latent space produces diverse place-field-like activity whose statistics (e.g., place field size) can be controlled by hyperparameters such as the network redundancy (ratio of network size to latent dimensionality; S1 Fig). We stress that this autoencoder perspective does not require thinking about the latent variables as the true inputs and outputs of the hippocampus, but instead serves as a useful abstract model of hippocampal representations consistent with internally-generated (e.g., attractor) dynamics or other computations (see Discussion).

**Three types of remapping.** Now that we have established how environmental variables map to neural activity, we can consider how this activity remaps across environments. But what exactly is remapping? Experimentally, remapping refers to the observation that spatial position is coded by a distinct sequence of neural activity in each environment. We will make this more precise by defining remapping across any two environments, *A* and *B*, as the case in which $\mathbf{r}^A(\mathbf{p}) \neq \mathbf{r}^B(\mathbf{p})$ for one or more spatial positions $\mathbf{p}$ (Methods 1.2). This broad definition not only includes complete and partial remapping, but also any other more subtle variations in neural activity (see Discussion).

To see how firing-rate maps may change, we will first restate the pseudo-linear encoding model from Eq 2 with an environment-specific index attributed to each variable that could be subject to change across environments (Methods 1.2). For a given environment *A*, we write,

$$\mathbf{r}^A(\mathbf{p}) = \mathbf{E}^A\mathbf{z}^A(\mathbf{p}) + \boldsymbol{\nu}^A(\mathbf{z}^A(\mathbf{p}); \mathbf{E}^A).$$

(4)

Now, with this equation, we can see that there are three possible ways for neural activity to remap as a function of environment: (i) changes to the linear encoder matrix, $\mathbf{E}^A$, (ii) changes to the latent variables themselves, $\mathbf{z}^A(\mathbf{p})$, and (iii) changes to the nonlinear null space function $\boldsymbol{\nu}^A(\cdot)$. We will consider each of these in turn.

The first case to consider is an environment-specific setting of the encoder matrix $\mathbf{E}^A$, which will change the mapping from latent space to neural space for each environment (Fig 2a). We note that this affects both the encoder and decoder, and we thus refer to it as *encoder-decoder remapping*. We can understand this case as causing shifts or rotations of the axes representing the latent variables, which can be visualized in neural state space or angle space (Fig 2b and 2c), causing a different sequence of place cell activations (cf. Fig 1b, right). This case is in fact analogous to the classic view of hippocampal remapping as a set of distinct maps [13,15], each requiring a unique decoder for spatial position.

The second case fixes the latent axes, and instead alters the latent trajectory through changes in the non-spatial, cognitive variables, $\mathbf{c}^A$, and their corresponding latent representation $\mathbf{z}_\mathbf{c}^A$ (Fig 2d). While changes to spatial position can seemingly lead to remapping if each environment uses a non-overlapping portion of space (e.g., $p \in [0, 1]$ for env. *A* and $p \in [1, 2]$ for env. *B*), this does not fall under our definition of remapping (Methods 1.2). As illustrated in Fig 2e, the latent trajectory then changes along the cognitive axis ('$E_c$') despite the positional axis ('$E_p$') remaining unchanged. This can also be visualized in angle space, where we

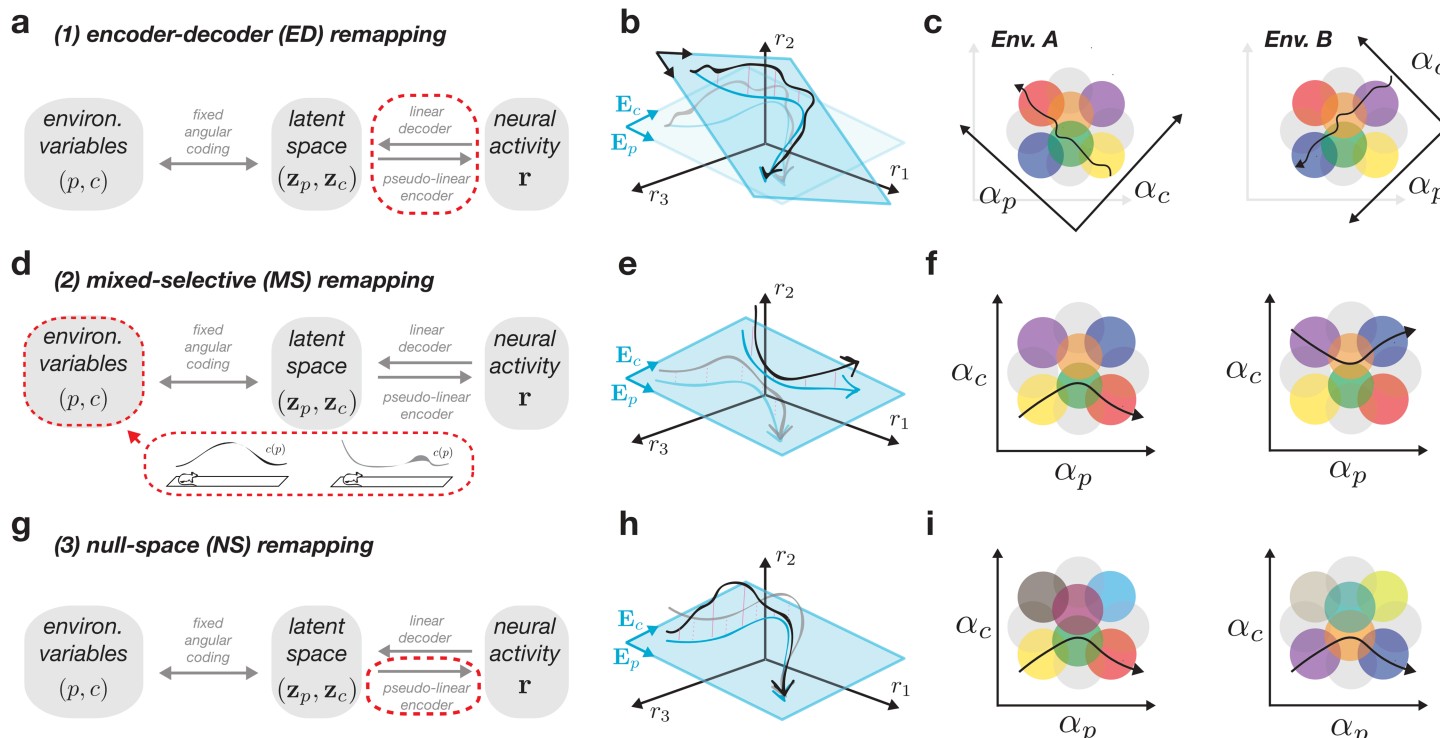

**Fig 2. Three types of remapping with linear decoders. a-c**: encoder-decoder (ED) remapping induces changes in the mapping between latent and neural spaces (**a**, dashed red), which can be visualized as rotating the latent space inside of a larger embedding space (**b**), or rotating the angle axes (**c**), such that the same latent trajectory passes through a different place cell sequence. **d-f**: mixed-selective (MS) remapping assumes changes to the cognitive environmental variables (**d**) and results in a different latent trajectory with a common positional readout (**e**), which passes through different place fields in angle space (**f**). **g-i**: null-space (NS) remapping features changes only to the nonlinear part of the pseudo-linear encoder but keeps decoding the same (**g**), resulting in different trajectories with the same underlying latent sequence (**h**), and thus can be seen as changing the place fields that support this trajectory (**i**).

see that the neurons' mixed selectivity dictates that they will only be active when the latent trajectory crosses the right conjunction of spatial and non-spatial information (Fig 1f; [53, 60]). We thus term this *mixed-selective remapping*. This type of remapping relates to more recent latent state space models of the hippocampus [31], and suggests that environmental differences in external (e.g., sensory) and internal (e.g., behavioral state) variables can lead to remapping even with a fixed decoder for spatial position (e.g., [61]).

Finally, in the third case, we consider the possibility for remapping to be explained solely through changes to the final, non-linear term $\nu^A(\cdot)$, which only affects encoding (Fig 2g). We note that the previous two types of remapping will, in general, also be accompanied by changes in this term (reflected in its dependence on both $\mathbf{z}^A(\mathbf{p})$ and $\mathbf{E}^A$; S2 Fig)—the difference here is that changes are restricted only to this term. We refer to this case as *null-space remapping*, as activity changes will be fully contained within the null space of the latent readouts (Fig 2h). Here, latent trajectories appear identical across environments, but the supporting neural firing fields will be modulated, indicating changes in tuning or excitability not captured by the linearly-decodable latent space (see Fig 2i). While the idea of null space activity has been discussed in other contexts (e.g., [62]), it has not been, to the best of our knowledge, related to remapping before (see Discussion).

We emphasize that our remapping theory only depends on one of the two model assumptions—linear decoding from neural activity to latent space. In this sense, the three

types of activity changes that we describe here will hold for any network with a linearly-decodable latent space (with the RNN model from Eq 3 serving as a particular concrete example). Our theory thus can serve as a useful framework to model neural variability in general across a variety of neural architectures (see Discussion). In contrast, the assumption of angular coding is rather a hippocampus-specific assumption which is sufficient to generate localized place-like firing fields, as well as an overall nonlinear relationship between the environmental variables and neural activity, despite linear decoding. Our focus for the remainder of the paper will be to characterize and demonstrate examples from each of these three remapping types, and to discuss how they relate to models and experiments from the literature.

**Encoder-decoder remapping.** We first examined encoder-decoder (ED) remapping, defined by changes in the mapping between latent and neural spaces via environment-specific encoding matrices, e.g., $\mathbf{E}^A$ and $\mathbf{E}^B$ for two environments $A$ and $B$ (Eq 4; cf. Fig 2a–2c). For simplicity, we temporarily omitted the non-spatial variables, leaving only the fixed positional latent variables ($\mathbf{z} = \mathbf{z_p}$), which are then $Z$-dimensional. In principle, each encoding matrix can then map the latent variables into any $Z$-dimensional subspace of $\mathbb{R}^N$. To explore these modulations in a systematic way, we focused on two concrete models: the multi-chart model and the grid realignment model, reflecting parallels to established theories in the literature [15,19].

In the multi-chart model, we constrain different encoding matrices, such as $\mathbf{E}^A$ and $\mathbf{E}^B$, to map the latents into a common subspace referred to as the "embedding space" (Methods 2.1). The dimensionality of this subspace limits how much the encoding matrices of different environments $A$ and $B$ can change. For instance, if the embedding space is $Z$-dimensional, then the encoding matrices can at most rotate in the same subspace; if the embedding space encompasses the full $N$-dimensional neural state space, then the encoding matrices can change in any possible way. A toy example of the multi-chart model for two linear tracks is shown in Fig 3a–3d. The results for larger networks representing 2-d square environments are shown in Fig 3e and 3f (see also S3 Fig).

We note two main characteristics of multi-chart ED remapping that match well with our theoretical intuitions. First, low-dimensional constraints on the embedding space induce a non-random, neighborhood-like structure in place field activity. We observed this in the toy model by the appearance of common groups of place fields across the two environments (Fig 3c and 3d, neurons 1-3), and in the scaled-up simulations through firing-rate overlap measures that were significantly larger than a random shuffle control (Fig 3f, top; Methods 4). This non-random overlap was consistent across parameters (S3a Fig). Second, ED remapping permits arbitrary affine transformations of spatial coordinates, allowing place fields to remap to any position across environments. This was clearly seen in the spatial localization of the place fields in the two linear track environments (Fig 3c and 3d), as well as in the scaled-up simulations with spatial correlation measures consistent with the random shuffle control (Fig 3f, bottom; and S3b Fig).

While the multi-chart model requires explicit environment-dependent changes in encoding weights, we next considered an analogous, but more plausible ED remapping model via grid realignment—a well-characterized phenomenon where phase shifts between grid modules in the entorhinal cortex can lead to different place cell activation patterns [19,63]. Rather than rotating the latent space in a higher-dimensional embedding space, in this model the latent space itself is made higher dimensional, now being composed of multiple grid modules. Environment-dependent changes in the encoding matrices are restricted to rotations within this space (Methods 2.1.2), which can be interpreted as each module's trajectory being rotated

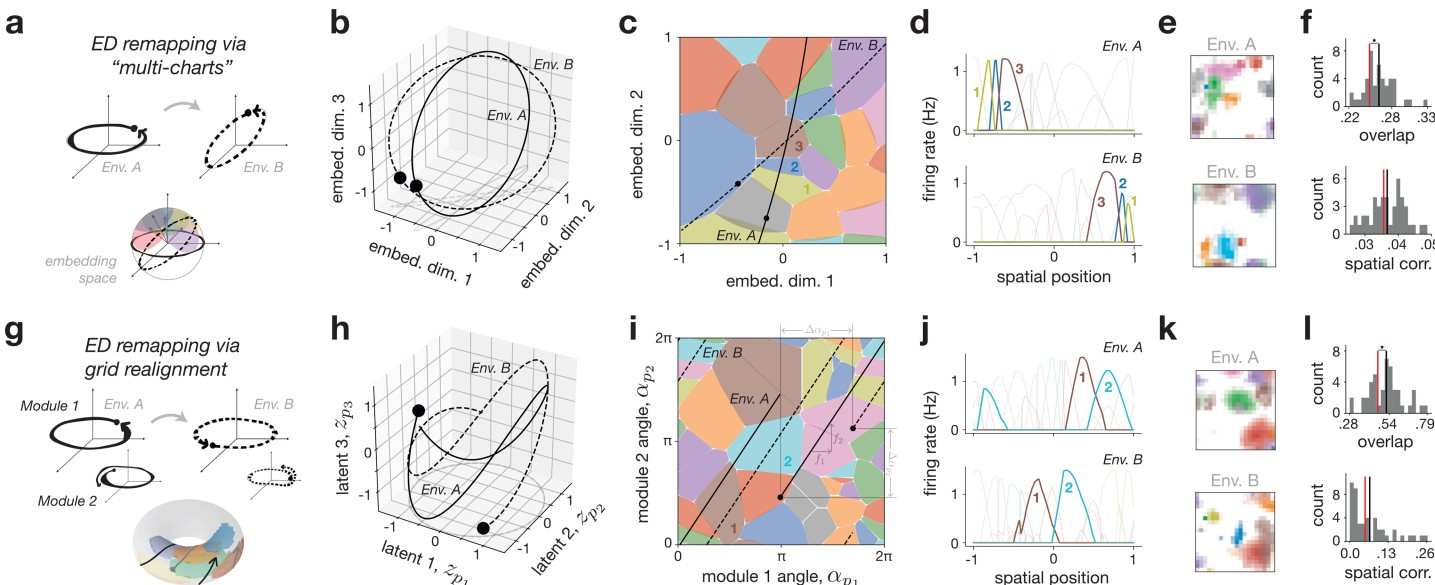

**Fig 3. Encoder-decoder (ED) remapping. a**: ED remapping via the "multi-chart" model. Considering a linear track, the 2-d latent space can be seen as being randomly rotated into a larger 3-d embedding space. Place cell remapping reflects the different alignment of neural tuning vectors with each environment's trajectory. **b,c,d**: A network of $N$ = 24 neurons is simulated with the setup from panel (**a**) for two environments $A$ and $B$; latent inputs follow rotated circular trajectories (**b**), and place fields are visualized in a 2-d projection of the 3-d embedding space (**c**) or as a function of position (**d**); three place cells are highlighted (1,2,3 in (**c**, **d**)). **e,f**: Example of scaled up multi-chart model with 2-d spatial position (4-d latent space) in a network of $N$ = 2048 neurons and a 128-dimensional embedding space (see S3 Fig for additional simulations), with rate maps from two example environments (**e**) and overlap and spatial correlation distributions (mean in black) compared with a shuffle control (red) (**f**); star indicates statistical significance (Methods 4). **g**: ED remapping via grid realignment. Considering a linear track with two grid modules (4-d latent space), the latent space can be seen as being realigned or rotated within itself in each environment, resulting in shifted trajectories on the very same toroidal latent manifold. **h,i,j**: A network of $N$ = 24 neurons is simulated with the setup from panel (**g**) and analogous to panels (**b,c,d**); latent inputs are shown in 3 of the 4 latent dimensions (**h**), and place fields are visualized in angle space (**i**) and as a function of position (**j**) with two place cells highlighted (1 and 2); realignment is due to phase changes in the starting position of each angle, and the common slope is due to the ratio of frequencies, $f_2/f_1$. **k,l**: Scaled up grid realignment model with 2-d spatial position in a network of $N$ = 96 neurons and 3 grid modules (12-d latent space), with rate maps from two example environments (**k**) and overlap and spatial corr. distributions (mean in black) compared with shuffle (red, **l**).

on its own axis (i.e., phase realigned; Fig 3g). We simulated both toy linear track and large-scale 2-d examples with two and three grid modules, respectively (Fig 3g–3l). As expected, the two key features of the multi-chart model were preserved. First, place fields exhibited a neighborhood structure (Fig 3i and 3j), leading to non-random overlap (Fig 3l, top), and second, place field location could arbitrarily change across environments (Fig 3i and 3j), leading to random spatial correlation (Fig 3l, bottom).

We made two additional observations about ED remapping. First, while network redundancy (the ratio of network size to embedding dimensionality) influenced place cell properties (S1 Fig) and affected raw overlap and spatial correlation values (S3 Fig), it did not alter the key signatures of ED remapping—namely, the non-random structure of overlap, and random structure of spatial correlation. In highly redundant networks, however, overlap could appear "seemingly random" (S3a Fig, left), and many more environments were needed to adequately sample and uncover the underlying population structure (S3a Fig, right; see Discussion). Second, in the special case of a full-dimensional embedding space (S4 Fig), the structured overlap of the multi-chart model vanished (S3a Fig), resembling the original model [15]. In contrast, the grid alignment model could not plausibly replicate this behavior since it would require an unreasonably large number of grid modules to match the latent dimensionality to the number of place cells.

**Mixed-selective remapping.** Next, we investigated mixed-selective (MS) remapping, which uses a shared latent space across all environments, but also includes cognitive (i.e., non-spatial) variables, $\mathbf{z_c^A(p)}$, that could change within and across each environment $A$ (cf. Fig 2d–2f). As above, here we focused on two particular representative examples of MS remapping, which we call space-feature coding and implicit-space coding, respectively, to illustrate the flexibility and constraints of this remapping class and its relation to recent models from the literature.

The space-feature coding model serves as the most generic implementation of MS remapping—environments share a common spatial latent trajectory, $\mathbf{z_p^A(p)} = \mathbf{z_p^B(p)} = \mathbf{z_p^C(p)}$ (referred to simply as $\mathbf{z_p(p)}$), but then are given different cognitive latent components, $\mathbf{z_c^A(p)} \neq \mathbf{z_c^B(p)} \neq \mathbf{z_c^C(p)}$ (Methods 2.2.1; Fig 4a). Neurons were tuned to be conjunctive to both sets of variables, similar to related models in the literature [32] (Methods 3.2.2). We simulated and visualized a toy example of this model for three linear track environments with a shared cognitive variable (Fig 4b–4d), along with scaled-up simulations for sets of 2-d square environments with different numbers of cognitive variables (Figs 4e and 4f and S5). The trajectories of the cognitive variables were generated from a Gaussian process (GP) whose variability we could also control (Methods 2.2.1).

We noted two main characteristics of MS remapping. One, similar to ED remapping, the shared low-dimensional latent space creates a neighborhood structure in place fields.

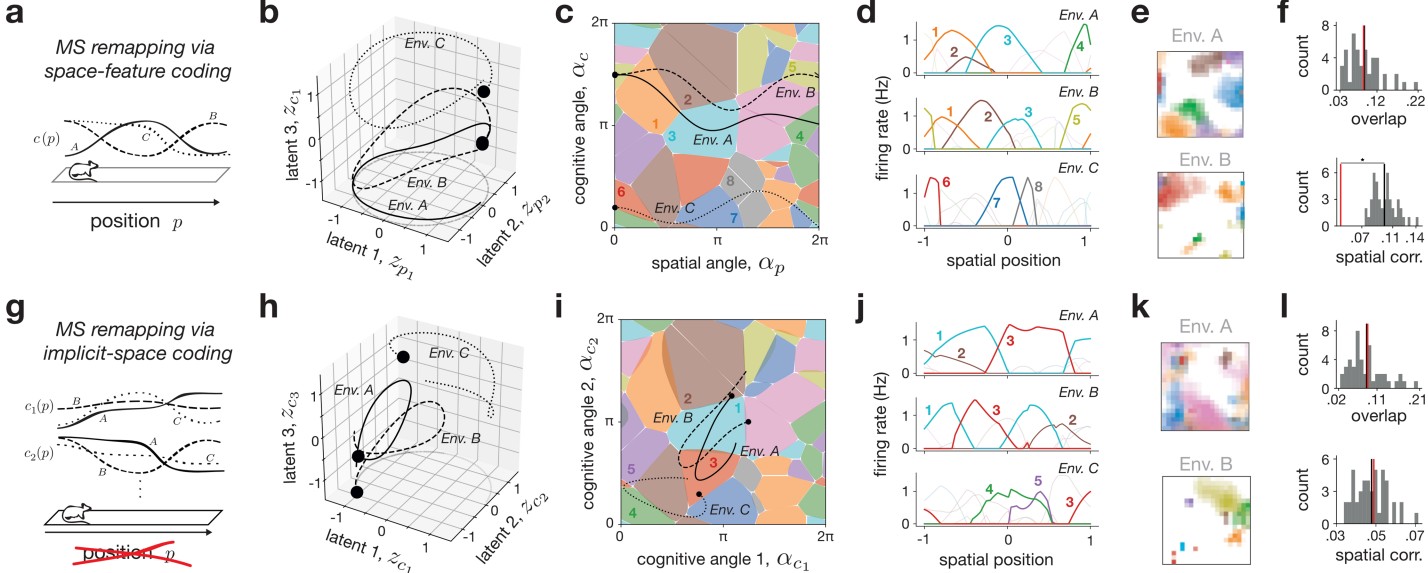

**Fig 4. Mixed-selective (MS) remapping. a**: MS remapping via space-feature coding; each environment features a shared position variable $p$, plus an environment-specific and position-dependent cognitive variable ($c^A(p)$ and $c^B(p)$). **b,c,d**: A network of $N = 32$ neurons is simulated using the space-feature coding setup from panel (**a**) for three environments; latent input trajectories visualized in 3 of the 4 latent dimensions shows circular positional trajectories (($z_{p_1}, z_{p_2}$) plane) with variability in the third, cognitive direction (**b**); trajectories and place fields are visualized in angle space (**c**) and as a function of position (**d**), with several place fields highlighted (1-8). **e,f**: Scaled up example of space-feature coding model with $N = 1024$ neurons and 32 environmental variables (including 2-d position; see S5 Fig for additional simulations), with example place field maps from two environments (**e**) and overlap and spatial correlation distributions (mean in black) compared with a shuffle control (red) (**f**). Star indicates statistical significance (Methods 4). **g**: MS remapping via implicit-space coding; similar to the position-dependent model, but with the spatial variables omitted so that only cognitive variables are represented (two shown for each environment, $c_1^A(p)$ and $c_2^A(p)$ for env. A). **h,i,j**: A network of $N = 32$ neurons is simulated using the implicit-space coding setup from panel (**g**) for three environments; latent input trajectories visualized in 3 of the 4 cognitive latent dimensions shows unconstrained trajectories (**h**), trajectories and place fields are visualized in angle space (**i**) and as a function of position (**j**), with several place fields highlighted (1-5). **k,l**: Scaled up example of implicit-space coding with $N = 1024$ neurons and 32 environmental variables (*excluding* 2-d position), with example rate maps from two environments (**k**) and overlap and spatial corr. distributions (mean in black) compared with shuffle (red, **l**).

This resulted in non-random, partial remapping in some pairs of environments (Fig 4b–4d, Envs. *A* & *B*), as well as overlap measures significantly exceeding that of shuffle controls across parameters (S5a Fig). As above, the extent to which this structure was evident depended on model parameters—specifically, networks with high GP variance (Fig 4b–4d, Envs. *A* & *C*), or a combination of high dimensionality, redundancy, and GP variance could exhibit "seemingly random" overlap (Figs 4e–4f and S5a and S5b, left) unless many environments were compared (S5a and S5b, right). Two, unlike ED remapping, the shared latent position across environments restricted the ability of individual place fields to shift arbitrarily with position. This again contributed to observations of partial remapping, and manifested in consistently non-random spatial correlation in the scaled-up simulations (Figs 4b–4f and S5c, S5d).

Inspired by some recent models in the literature [33,35,64], we then considered the implicit-space model, a special case of MS remapping that does not contain any explicit spatial variables (Methods 2.2.2). Instead, latent trajectories are composed solely of cognitive variables, and so each position **p** can map to arbitrary locations in latent space (Fig 4g and 4h). We simulated and visualized a toy example of three linear track environments now with a pair of cognitive variables (Fig 4h–4j), and a set of scaled-up simulations of 2-d square environments with higher-dimensional cognitive latent trajectories again generated from a GP (Fig 4k and 4l). The results situate the implicit-space model as a flexible intermediary between ED and space-feature MS remapping. When cognitive trajectories coincide, neural activity exhibits structure across environments similar to the partial remapping noted in the space-feature MS model (Fig 4h–4j, Envs. *A* & *B*). However, in the limit of random uncorrelated trajectories across environments, the implicit space model resembles ED remapping, exhibiting seemingly random overlap, as well as truly random spatial correlation (Fig 4h–4j, Envs. *A* & *C* and Fig 4l).

**Null-space remapping.** Considering the pseudo-linear encoder in Eq 4, we see that ED and MS remapping will not only cause changes to the linear term, but also to the nonlinear null space term, due to its dependence on encoding weights and latent trajectories (S2 Fig). In null-space (NS) remapping, however, we propose a form of remapping where all activity changes are confined to the null space term. This can be visualized as follows. Let us consider two place cells that share the same spatial tuning preference, and which are connected through lateral inhibition (Fig 5a and 5b). Given the linear decoder assumption, we can visualize the latent readout in the two-dimensional neural state space on the diagonal between the two axes (Fig 5a, blue), along with an orthogonal null space axis (Fig 5a, pink; [65]). Then, assuming inhibitory competition between the two neurons, we can see that there are three qualitatively different activity regimes yielding the same latent output—either one of the two neurons outcompetes the other and fires alone, or the two jointly fire together (Fig 5b). In fact, the recurrent network model that we have employed here has the exact lateral-inhibitory structure needed for this effect ([57,66,67]; see Discussion). While this phenomenon has previously been suggested as a cause for trial-to-trial variability, we here suggest that it can also be seen as a type of "remapping" under certain circumstances.

The situation with several place fields is illustrated in Fig 5c. Here, three trajectories evolve in a similar way along the latent spatial dimension, but evolve differently along the "hidden" null space dimension, which causes different place cells to become activated. This type of remapping is thereby similar to MS remapping (Fig 4c), with the notable difference that here the hidden null space dimension carries no information.

A plausible cause of NS remapping is variability in the set of neurons that actively participate in the hippocampal map at any given time, which we refer to as "participation

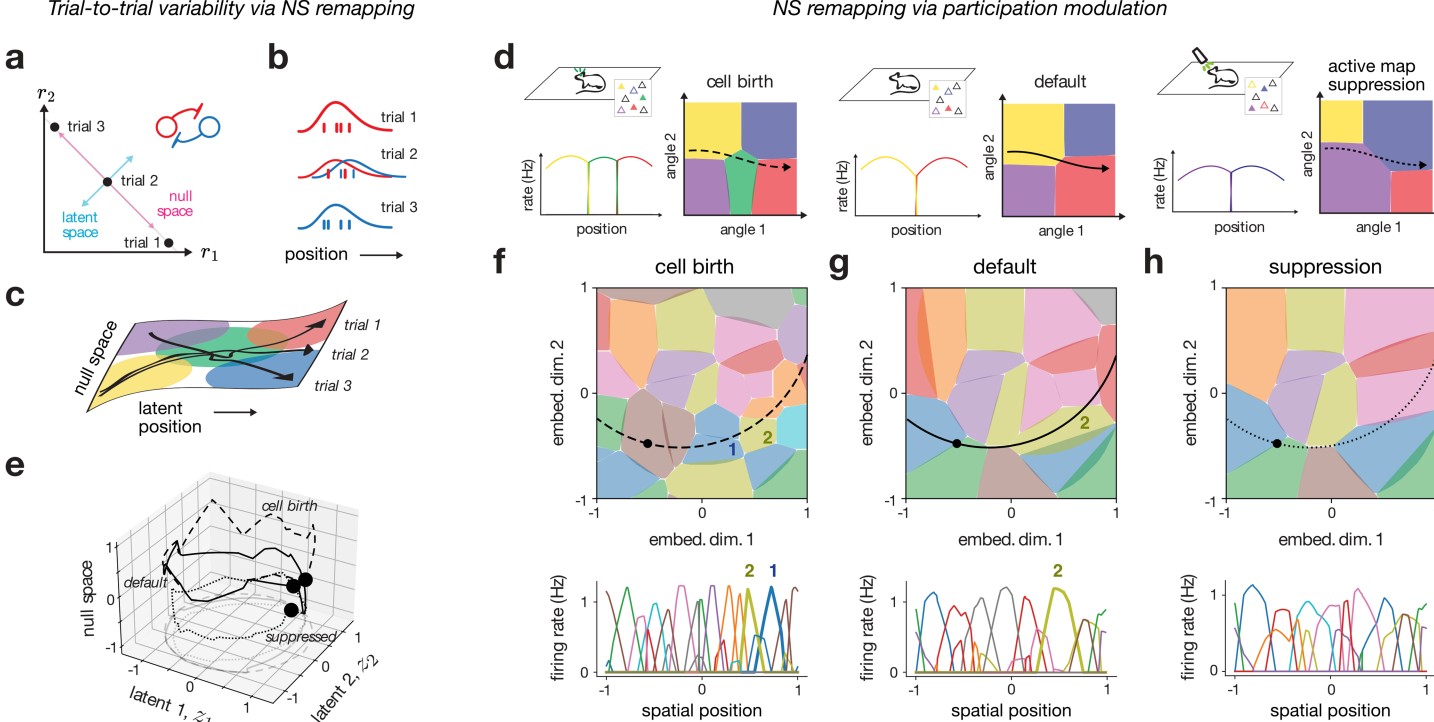

**Fig 5. Null-space (NS) remapping. a**: Schematic of how variability in a null space direction may cause place fields to be modulated, resulting in trial-to-trial variability (cf. Fig 2i). **b,c**: In two-dimensional ($r_1, r_2$) neural space, a single latent variable (light blue) is coded, with an orthogonal null space direction (pink); a competitive architecture with lateral inhibition results in trials with distinct activity for neuron 1 (red) and neuron 2 (blue). **d**: Illustration of experimental setup, competition in angle space, and resulting place fields; a default map (center, red and yellow neurons) is modulated by the participation of a new neuron (left, green neuron) or suppression and replacement with an alternative map (right, purple and blue). **e,f,g, h**: Simulated example of the setup from panel (**d**), using a single environment of the multi-chart ED model from Fig 3b–3d ($N = 48$ neurons; see S6 Fig for additional simulations). Circular positional trajectories for default (solid), cell birth (dashed) and supressed (dotted) are plotted with the maximum-variance null space direction (**e**). The cell birth (**f**), default (**g**), and supressed (**h**) place field maps are shown in angle space (top) and as a function of position (bottom). Two neurons, 1 and 2, are highlighted.

modulation", and which can be controlled by neural excitability. We note that such excitability shifts will generally not be tied to particular environments per se, but rather to particular timescales and manipulations. Such systematic changes could occur relatively rapidly (trial-to-trial), or over much slower timescales (e.g., slow metabolic changes, cell birth/death, experimental ablation), and thereby modulate which subset of neurons actively take part in the representation.

We exemplified and visualized NS remapping through toy schematics (Fig 5d) and simulations (Figs 5e–5h and S6), in which a "default" map (Fig 5d, middle; and Fig 5g) is modulated by either adding neurons ("cell birth"; Fig 5d, left; and Fig 5f), or removing neurons ("suppressed"; Fig 5d, right; and Fig 5h) from the representation. Adding neurons caused them to out-compete existing neurons (Fig 5d, left; and Fig 5f), resulting in a change in activity and subtle modulations of the spatial preference of particular neurons. In the limit that all active neurons were suppressed, an entirely different map appeared of previously inactive neurons that compensated for the loss (Fig 5d, right; and Fig 5h), echoing a recent experimental protocol [68]. The latent trajectories remained relatively stable, with changes restricted primarily to the null space (Fig 5e). Furthermore, these changes manifested as subtle, non-random overlap and spatial correlation, which could appear random if the modulation was made dense enough (S6a and S6b Fig). NS remapping thus yields structured, non-random overlap and

spatial correlation similar to MS remapping, despite distinct underlying causes (Methods 3.2.3; Discussion).

**Case study: Reward-modulated place fields.** To illustrate how our remapping framework can be applied in practice, we modeled a task where an animal navigates a linear track with a reward or goal that shifts location (Fig 6a). Goal-modulated hippocampal responses have been widely studied [4,69], revealing both "feature-in-place" cells, tuned to the conjunction of spatial position and reward [70,71], as well as pure reward cells, which fire irrespective of location [24,26]. These findings have fueled contrasting theories, with one viewing the hippocampus as primarily spatial [4], and the other as representing a more general latent space [31,72]. Here, we ask: what kinds of single-neuron representations are consistent with these views, and can they be distinguished with our framework?

Using the space-feature MS remapping model (cf. Fig 4a–4e), we simulated a latent space composed of spatial position and a single cognitive feature representing the reward location. We then compared two types of neural tuning that affect how neurons "tile" the latent space (Fig 6b), and may reflect different assumptions about input organization from entorhinal cortex (EC) [32,73,74]. In the conjunctive model, neurons respond only when both spatial and reward-related inputs match their tuning (e.g., reflecting separate normalized input streams from medial and lateral EC, respectively; Fig 6b, left). In the pure & mixed model, neurons are not constrained to strictly be conjunctive, and so in addition to some neurons with mixed tuning, others exhibit purely spatial or reward tuning (Fig 6b, right, vertical and horizontal colored ellipses; Methods 3.2.2).

Simulations of two reward conditions revealed three key observations (Fig 6c–6f). First, both models showed stable place fields outside the reward zones and partial remapping with more place fields inside the reward zones—the phenomenon of denser firing near rewards is consistent with experimental findings [75] and attributable in our model to extended latent trajectories around the goal. Second, in the conjunctive model, different neurons were active at each reward location, consistent with feature-in-place coding (Fig 6c and 6d, neurons 2 and 4). Third, in the pure & mixed model, the same reward-tuned neurons remapped to the new location, resembling pure reward cells (Fig 6e and 6f; neurons 2 and 3). These results lead to two conclusions. First, our remapping framework can account for a large diversity of experimentally-observed feature selectivity profiles through appropriate assumptions on the

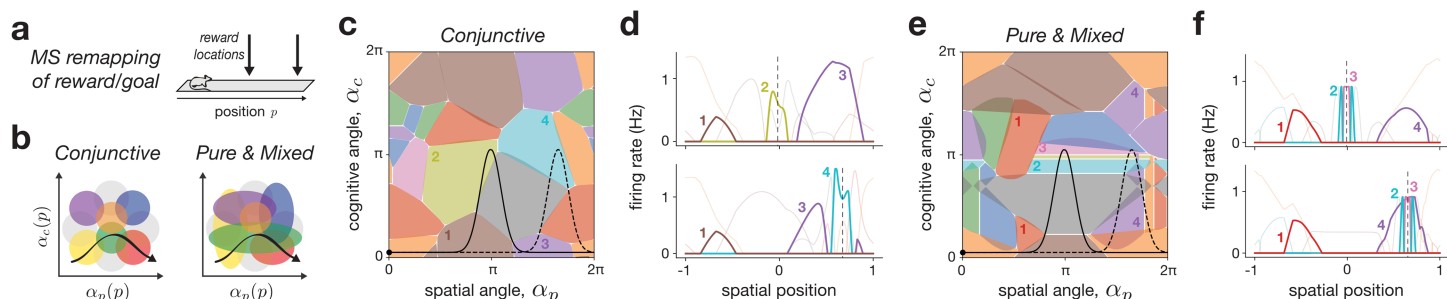

**Fig 6. Two types of reward coding with mixed-selective remapping. a**: Experimental setup, in which an animal navigates through a linear track environment, with two distinct reward locations. **b**: Two distributions of neural tuning, in which neurons are tuned with equal weighting towards position and reward intensity (conjunctive, left), or in which neurons are tuned on a spectrum from purely selective to mixed-selective (pure & mixed, right) (Methods 3.2.2). **c,d**: MS remapping with conjunctive selectivity in a network of $N = 16$ neurons and 4-d latent space; two latent trajectories follow the reward location, overlapping with different firing fields in angle space (**c**) and resulting in a different sequence of place cell activity (**d**, reward location is marked by dashed vertical line, and four neurons highlighted 1-4). **e,f**: Same as panels (**c,d**) but with a combination of pure and mixed tuning, resulting in some neurons with pure selectivity for either position or reward, and others conjunctive (four neurons highlighted 1-4).

population geometry in feature space. Second, the fact that both response types are compatible with a spatial cognitive map suggests caution when interpreting single-neuron properties as evidence for broader hippocampal function.

## Discussion

In this work, we have presented a unifying view of hippocampal place field remapping from the perspectives of neural coding and population geometry. While many previous studies have employed a population-level perspective on spatial representations and their variability [6,44,46,47,56,90–92], our work introduces a more general and principled framework for remapping—one that unifies a wide range of empirical observations under a single, intuitive theoretical model. By assuming a linearly-decodable latent state space, we demonstrated three mechanisms—encoder-decoder (ED), mixed-selective (MS), and null-space (NS) remapping—that each may underlie neural activity changes, including complete and partial remapping. Rather than privileging one mechanism, we contend that all three, either in isolation or in combination, are likely accurate depictions of activity changes in the hippocampus under different experimental conditions. Our modeling perspective can thus serve as a testbed to explore how various experimental settings and manipulations impact population codes and remapping statistics.

We summarize the key characteristics of each remapping type in Table 1, focusing on overlap and spatial correlation, along with connections to various experimental and computational studies from the literature (Methods 5). Concerning overlap, we found that full-dimensional ED remapping was the only mechanism that lacked local neighborhood structure and thereby exhibited fully random overlap. Other models showed non-random overlap. However, depending on parameters or modulation strategies, this non-random overlap could become weak enough so as to appear random if data sets are not large enough ('seemingly random'). Concerning spatial correlation, we found it a more reliable metric to distinguish mechanisms. Indeed, ED remapping generally leads to random spatial correlation, since it

**Table 1. Summary and distinguishing fingerprints of each remapping type and their relationship to published experimental studies and computational models (Methods 5 for more details).** Note that the final column, "active map suppression" refers to remapping with a non-overlapping population of neurons, and thus overlap and spatial correlation measures cannot strictly be measured.

| | Encoder-Decoder (ED) | | Mixed-Selective (MS) | | Null-Space (NS) | |
|---|---|---|---|---|---|---|
| | full-D multi-chart (S4 Fig) | multi-chart & grid (Figs 3 and S3) | implicit space (Fig 4g–4l) | space-feature (Figs 4a–4f and S5) | participation modulation (Fig 5f and S6) | active map suppression (Figs 5h and S6) |
| Random overlap | yes | | | | | |
| Seemingly-random overlap | | high redundancy | high redundancy and high variance | | dense modulation | |
| Non-random overlap | | low redundancy | low redundancy or low variance | | sparse modulation | |
| Random S.C. | yes | yes | high variance | | | |
| Non-random S.C. | | | low variance | yes | yes | |
| Experiments | [12,18–20,47,76] | | [24,25,46,61,77–79] | | [80–83] | [68] |
| Models | [15,16] | [32,39,63,84] | [33–38,64] | [32,85,86] | [59,87–89] | |

permits arbitrary restructuring of the latent space, while MS and NS remapping lead to structured correlations, since they preserve shared positional axes. An exception is implicit-space MS, which can approximate random spatial correlation.

Our framework also re-contextualizes various remapping findings and theories (Methods 5). The two ED implementations linked classical multi-chart [15] and grid realignment [19,63] models in analogous low-dimensional population spaces. While multi-chart latent space rotations may have some plausibility (e.g., [93]), they also serve as an abstraction for how grid modules and their realignment effectively tile high-dimensional space [94,95]. Such intuition could also apply to other types of spatial inputs like boundary vector cells [96,97], and may also help to explain how the diverse sensory and cognitive variables associated to MS remapping can lead to highly non-overlapping representations [32,33]. Additionally, as explored for the case of reward coding (cf. Fig 6), the interaction of a latent trajectory with the nature of the place cells' mixed-selectivity [60] can explain various experimental findings, including splitter cells [77,78], and contextual [79,98] and behavioral modulation [61].

We introduced a novel mechanism—null-space (NS) remapping—to explain changes that preserve the latent representation but modify which neurons contribute to it. NS remapping accounts for trial-to-trial variability [56,99], slower changes in cell participation [81,83], and experimental perturbations [68]. It supports a view of the hippocampus as having competitive dynamics [100], where neurons compete not just between maps but also within the same active map. NS remapping also relates to mechanisms proposed in working memory [101] and motor control [62]. Functionally, NS and MS remapping can appear similar and indistinguishable, as both involve changes orthogonal to latent position. The key distinction is that MS remapping reflects modulations along known or controlled variables, while NS remapping occurs along uncontrolled or hidden dimensions (cf. Figs 4b and 5e). This sets ED remapping apart as the only mechanism that truly alters the cognitive map for space, hence the differences highlighted in Table 1. The fixed positional readout for MS and NS remapping may help to explain experiments in which remapping occurs with unchanging readouts or behavioral performance [82,102].

Our framework features two important assumptions. First, angular coding serves as a convenient and interpretable latent embedding. This choice induces constant population activity (as observed in the hippocampus; [103]), localized firing fields [52], and permits multi-field tuning [27,28], without enforcing periodicity (Methods 1.3). Second, we assume a linear decoder from neural to latent space, which is necessary to derive the three remapping mechanisms. This choice is well motivated [53,60,104], and does not imply that space can be linearly decoded from neural activity (due to the nonlinear angle coding), nor that neural activity linearly encodes the latent variables. In summary: angular coding is sufficient (but not necessary) for generating place-fields; linear decoding is necessary for the remapping framework, independent of angular coding.

An important limitation of our model is its simplicity: we abstracted the hippocampus as an autoencoder, without modeling internal dynamics, coupling with grid cells, nor any unsupervised or task-based learning [32,33,35,38,86,105–109]. This was intentional—we set out to understand hippocampal representations in the most abstract sense, as modeling a particular learning process or task would have narrowed the generality of the results. That said, our results can neither speak to the utility nor plausibility of particular representations. Future extensions could incorporate internally-generated dynamics [31], or discrete map switching [110,111]. Notably, the architecture we employed here has already been applied to model path integration [57], attractor memories [112], and other dynamics [113,114], which could be explored in the future.

We chose to limit our study to deterministic mappings from position to activity, but variability is a central features of hippocampal coding [6,46,56,99,115]. Our framework could accommodate this in multiple ways, such as through stochasticity of the environment variables [46,116], or spike-based internally-generated trial-to-trial variability ([67]; cf. Fig 5a–5c), which could open up our framework to the analysis of shorter timescale co-firing patterns [6,47,117,118]. One additional type of variability is the drift or instability of place cell activity over time [81], which has been linked to remapping elsewhere [31,39]. From the perspective of our theoretical framework, this drift can be attributed to each of the three remapping types—learning or plasticity-induced modifications may change the latent mapping (ED remapping) [39,76,119], changes in behavioral or cognitive variables may seemingly result in drift (MS remapping) [61,85], or non-coding excitability changes or cell birth/death may cause null-space (NS) remapping [83,89].

Lastly, our work is more general than remapping, and describes how mixed selectivity and linear decoding lead to an interpretable understanding of neural representations and how they vary [53]. Many of the theoretical results were obtained without any explicit assumptions of place or spatial tuning (Methods 1). As such, this framework could be useful in characterizing geometry and variability in other areas (e.g., [120,121]), contributing to ongoing research in neural manifolds and population geometry [41,42,122,123]. More generally, it offers a principled perspective on how low-dimensional variables can be robustly and flexibly embedded in high-dimensional neural activity.

## Methods

The Methods is divided into five sections. In Sect 1 (Theory), we describe the constrained model that maps environmental variables to neural activity through a latent space (Fig 1), the three types of remapping (Fig 2), and other specific details about angular coding. In Sect 2 (Examples of remapping), we describe the specific examples of each remapping type and modeling choices that went into each one. In Sect 3 (Simulations) and Sect 4 (Data Analysis), we describe the details of the RNN simulations and how we analyzed the data from these simulations. Lastly, in Sect 5 (Summary table), we provide additional details about the experimental and computational references provided in Table 2.

### 1. Theory

We begin from the hypothesis that the neural activity of $N$ neurons in the hippocampus, $\mathbf{r} \in \mathbb{R}^N$, represents spatial position, $\mathbf{p} \in \mathbb{R}^P$, along with other internal and external cognitive variables, $\mathbf{c} \in \mathbb{R}^C$, which together we call environmental variables. We are thus interested in the following neural coding problem (Fig 1a):

$$\underbrace{(\mathbf{p}, \mathbf{c})}_{\substack{\text{environmental variables} \\ \mathbb{R}^{P+C}}} \quad \longleftrightarrow \quad \underbrace{\mathbf{r}}_{\substack{\text{neural activity} \\ \mathbb{R}^N}} \tag{5}$$

To constrain this model and arrive at some concrete conclusions about remapping, we introduce the latent variables $\mathbf{z} = (\mathbf{z_p}, \mathbf{z_c}) \in \mathbb{R}^Z$ as an intermediate representation. Then, we consider the transformation from $(\mathbf{p}, \mathbf{c})$ to the latent variables $\mathbf{z} = (\mathbf{z_p}, \mathbf{z_c})$ and from the latent variables $\mathbf{z}$ to the rates $\mathbf{r}$. To keep our model tractable and aligned with common experimental remapping paradigms, we assume a deterministic mapping from position, $\mathbf{p}$, to all other

**Table 2. Glossary of mathematical symbols.**

| Variable(s) | Properties | Description |
|---|---|---|
| $P$ | | Number of position variables |
| $C$ | | Number of cognitive variables |
| $m$ | $\begin{cases} m > 1 & \text{for grid realignment ED} \\ m = 1 & \text{otherwise} \end{cases}$ | Number of grid modules |
| $Z$ | $Z = 2mP + 2C \leq N$ | Latent space dimensionality |
| $Y$ | $\begin{cases} Z \leq Y \leq N & \text{for multi-chart ED} \\ Y = Z & \text{otherwise} \end{cases}$ | Embedding space dimensionality |
| $N$ | | Network size |
| $\mathbf{p}, \mathbf{c}$ | $[-1,1]^P, [-1,1]^C$ | Environmental variables (position and cognitive) |
| $\boldsymbol{\alpha}_p, \boldsymbol{\alpha}_c$ | $[0, 2\pi]^{mP}, [0, 2\pi]^C$ | Angular variables (position and cognitive) |
| $\mathbf{z} = (\mathbf{z_p}, \mathbf{z_c})$ | $\mathbb{R}^{2mP}, \mathbb{R}^{2C}$ | Latent variables (position and cognitive) |
| $\mathbf{y}$ | $\mathbb{R}^Y$ | Embedding variables |
| $\mathbf{r}$ | $\mathbb{R}^N$ | Neural activity |
| $\mathbf{D}, \mathbf{D_y}$ | $\mathbb{R}^{Z \times N}$ | Decoding matrix (w.r.t. latent or embedding spaces) |
| $\mathbf{E}, \mathbf{E_y}$ | $\mathbb{R}^{N \times Z}; \mathbf{E} = \mathbf{D}^+$ | Encoding matrix (w.r.t. latent or embedding spaces) |
| $\boldsymbol{\nu}(\mathbf{z})$ | $\mathbb{R}^Z \to \mathbb{R}^N$ | Nonlinear null space function for encoding model |
| $\mathbf{R}$ | $\mathbb{R}^{Y \times Z}$ | Linear mapping between latent and embedding spaces |
| $A, B, C$ | | Denotes a particular environment (e.g., $\mathbf{z}^A, \mathbf{r}^A, \mathbf{E}^A$) |

variables in the model. We can thus write the general coding model as (Fig 1c):

$$\underbrace{(\mathbf{p}, \mathbf{c}(\mathbf{p}))}_{\substack{\text{environmental variables} \\ \mathbb{R}^{P+C}}} \quad \longleftrightarrow \quad \underbrace{\mathbf{z}(\mathbf{p}) = (\mathbf{z_p}(\mathbf{p}), \mathbf{z_c}(\mathbf{p}))}_{\substack{\text{latent variables} \\ \mathbb{R}^Z}} \quad \longleftrightarrow \quad \underbrace{\mathbf{r}(\mathbf{p})}_{\substack{\text{neural activity} \\ \mathbb{R}^N}} \tag{6}$$

This dependence on $\mathbf{p}$ implies that all environments are treated as unchanging and fixed—therefore any variation, no matter how subtle, is seen as a "new" environment (see Methods 1.2 below). At times we will omit writing the explicit dependence of all variables on $\mathbf{p}$ for brevity and readability in the notation.

Below, we will describe the motivation and formulation of each mapping in Eq 6, following the visual schematic description in Fig 1c–1f. In contrast to the main text, we will begin with the $\mathbf{z} \leftrightarrow \mathbf{r}$ mapping, as the choice of linear decoding stands as the central assumption of our theory. Following this, we will then show how the inverse of this mapping, the pseudo-linear encoder, motivates the three types of remapping that we describe. Finally, we will then present the angle coding that describes the $(\mathbf{p}, \mathbf{c}) \leftrightarrow \mathbf{z}$ mapping. This ordering reflects the fact that angular coding is not necessary for our theory on the three types of remapping, but rather serves as a concrete modeling choice to generate place-field-like firing patterns.

**1.1. Latent representation: Mapping between latent variables and neural activity. Linear decoding and pseudo-linear encoding:** We assume a linear decoder mapping from neural activity to latent space of the form

$$\underbrace{\mathbf{r}}_{\substack{\text{neural activity} \\ \mathbb{R}^N}} \quad \mapsto \quad \underbrace{\hat{\mathbf{z}} = \mathbf{Dr}}_{\substack{\text{latent variables} \\ \mathbb{R}^Z}} \tag{7}$$

where $\mathbf{D} \in \mathbb{R}^{Z \times N}$ is the decoding matrix and $\hat{\mathbf{z}}$ represents estimates of the latent variables $\mathbf{z}$. Note that throughout this work we assume accurate de- and encoding of the latent variables

in the network activity ($\hat{\mathbf{z}} \approx \mathbf{z}$) and thus in Eq 1 we simply use $\mathbf{z}$. To obtain a functional form for an encoding model consistent with linear decoding, we consider an inversion of the linear decoding model, which can be written as

$$\underbrace{\mathbf{z}}_{\substack{\text{latent variables} \\ \mathbb{R}^Z}} \quad \mapsto \quad \underbrace{\mathbf{r} = \mathbf{Ez} + \boldsymbol{\nu}(\mathbf{z}; \mathbf{D})}_{\substack{\text{neural activity} \\ \mathbb{R}^N}} \tag{8}$$

where $\mathbf{E} = \mathbf{D}^+ \in \mathbb{R}^{N \times Z}$ is the right pseudo-inverse of the decoding matrix, and $\boldsymbol{\nu}(\cdot) \in \mathbb{R}^Z \to \mathbb{R}^N$ is a nonlinear function in the null space of $\mathbf{D}$. We refer to Eq 8 as a pseudo-linear encoder, since the first term specifies a linear component to the encoding, which can then be made highly nonlinear in practice due to the second term. To see that the pseudo-linear encoder implies a linear decoder, one can simply multiply Eq 8 by $\mathbf{D}$ to recover Eq 6, noting that $\mathbf{D}\mathbf{D}^+ = \mathbf{Id}_Z$ and $\mathbf{D}\boldsymbol{\nu}(\mathbf{z}) = \mathbf{0}$. Since $\boldsymbol{\nu}$ is a function of $\mathbf{z}$, it can change within each environment as a function of latent state, but also across environments for the case of null-space remapping. Furthermore, $\boldsymbol{\nu}$ is in the null space of, and thus parametrized by, the decoder $\mathbf{D}$. Using a slight abuse of notation, we write this dependence in Eq 2 of the main text with respect to $\mathbf{E}$ (instead of $\mathbf{D}$) to highlight the dependence of $\boldsymbol{\nu}$ on each component of the linear term in Eqs 2 & 8.

**Key considerations and alternative models:** The encoding model is the key component that enables us to study remapping. Importantly, the pseudo-linear encoder as written above (Eq 8) is the most general encoding model consistent with linear decoding—the nonlinear term $\boldsymbol{\nu}(\cdot)$ can be any function provided it is in the null space of $\mathbf{D}$. Thus, as we show in the following section, linear decoding is the sole assumption needed to derive the three types of remapping studied in this work. However, Eq 8 is too general to serve as a mechanistic network model of hippocampal representations for example simulations—this is why later, in Sect 3.2, we introduce an explicit RNN model consistent with Eqs 7 and 8 which also fully specifies the null space.

We chose linear decoding for reasons of simplicity, biological plausibility, and interpretability in visualizations [53,54,124]. Not only is such a linear readout similar to how a downstream neuron might plausibly read information out of a network, but it also follows linear dimensionality reduction methods like principal components analysis (PCA) which are commonly used in population analysis [125]. However, rather than starting with a decoding model that can be inverted to form an encoding model, an alternative would have been to specify an explicit encoding model. Generally speaking, network-level encoding models are difficult to specify, as encoding is an under-constrained problem (more neurons than latent variables), and it is generally agreed that networks of the brain non-linearly encode stimuli [54]. That said, two simple alternative options would have been to (1) use a feedforward linear-nonlinear model for each neuron [49,64,96], thereby ignoring recurrent interactions or (2) use a black-box autoencoder framework from machine learning (e.g., [126,127]), thereby risking a loss of interpretability. Some latent space models of hippocampal place cells use an objective akin to autoencoding [33,36–38], but they arrive at particular solutions whose generality is difficult to ascertain.

**1.2. Three types of remapping.** Before making any assumptions about the mapping from environmental variables to latent variables (($\mathbf{p}, \mathbf{c}) \leftrightarrow \mathbf{z}$), we can already use the pseudo-linear encoder formulation from Eq 8 to spell out the three types of remapping. For the moment, we assume an arbitrary one-to-one mapping $\mathbf{p} \leftrightarrow \mathbf{z}(\mathbf{p})$ which will be expanded upon in the following section. To model remapping, we repeat the definition that we state in the main text, which is that for two environments $A$ and $B$, we have

$$\exists \mathbf{p} : \mathbf{r}^A(\mathbf{p}) \neq \mathbf{r}^B(\mathbf{p}), \tag{9}$$

which means that the firing-rate maps for the two environments are distinct in at least one location. There are two important properties to note about this definition, as we briefly mention in the main text. One, all environments must share the same positional variables—this accounts for the possibility that environments may have different sizes or use different ranges of positions. If two environments utilize non-overlapping positional variables, this will indeed result in different neural activity in our model, but we do not consider this case. Not only does it not fall under our definition of remapping (since it is still consistent with the same map between neural activity and position) but it scales poorly and does not appear to be a plausible model of spatial coding across environments. The second important property about our remapping definition is that its generality means that any activity changes are consistent with remapping—from a complete map change as in random global remapping, to the subtlest of rate remapping, even to trial-to-trial variability around a set of otherwise stable place fields (see Discussion). This stems from the fact that remapping, as commonly used, is not a well-defined phenomenon (for a nice discussion of this, see Sanders et al. 2020 [39]). In an attempt to make it more strict, our definition could be refined to reflect trial-averaged rate maps, or by thresholding small rate changes, but would leave our theoretical results qualitatively intact.

Using the definition of remapping above, we thus consider the transformation from $\mathbf{z} = (\mathbf{z_p}, \mathbf{z_c})$ to rates $\mathbf{r}$ to be variable across environments by adding an environment-specific index to all possible variables (with the exception of $\mathbf{p}$ which we have already argued should be shared across environments). In doing so, we obtain an environment-specific pseudo-linear encoder equation

$$\mathbf{r}^A(\mathbf{p}) = \mathbf{E}^A \mathbf{z}^A(\mathbf{p}) + \boldsymbol{\nu}^A(\mathbf{z}^A(\mathbf{p}), \mathbf{E}^A). \tag{10}$$

From this equation, we can now see that there are three ways of changing the firing rate map with the environment, by changing the encoder, $\mathbf{E}^A$, the (cognitive part) of the latents, $\mathbf{z}^A = (\mathbf{z_p}, \mathbf{z_c}^A)$, or the null space non-linearity, $\boldsymbol{\nu}^A$. This leads to the three distinct remapping types described in the main text.

**1.3. Internal representation: Mapping between environmental and latent variables. Motivation:** Neural population activity may exhibit different extrinsic and intrinsic geometries [41,123]. In the context of spatial representations, this relates to the fact that euclidean space in the external world need not be represented by the same geometry in neural activity. There are two main reasons that motivate a more nonlinear, curved mapping. First, neural codes should be energy efficient, and are often modeled with explicit or implicit activity regularization or normalization which confines activity within or onto a hypersphere around the origin [128,129]. From this perspective, it is the direction rather than the magnitude that determines the representation. Second, place field activity is often confined to one or multiple localized areas of space. This motivates a curved trajectory that only locally aligns with each neuron's preferred tuning vector, which in ideal abstraction becomes a periodic or toroidal map [130]. In contrast, a purely linear representation is only consistent with monotonic tuning curves. Therefore, we choose an angular code as a simplistic model that accounts for both energy-efficient coding and localized place-field-like tuning. We note that this mapping is fixed across environments, with the idea being that the geometry of the internal representation of space should not be the main factor that accounts for remapping. However, evidence for internal estimates of spatial geometry [131] could motivate this assumption to be relaxed in future work.

**Angular coding:** For the fixed non-linear mapping between the positional variables $\mathbf{p} = (p_1, \ldots, p_P)$ (normalized as $\forall i,\ p_i \in [-1, 1]$) and the latent variables, $\mathbf{z_p}$, we choose an angular encoding of the form

Angular encoding:

$$\underbrace{p_i}_{\substack{\text{position} \\ [-1, 1]}} \quad \mapsto \quad \underbrace{\alpha_i = \pi(p_i + 1)}_{\substack{\text{angle} \\ [0, 2\pi]}} \quad \mapsto \quad \underbrace{(z_{p_{2i}}, z_{p_{2i+1}}) = (\cos(\alpha_i), \sin(\alpha_i))}_{\substack{\text{latent variables} \\ S_1 \subset \mathbb{R}^2}} \tag{11}$$

Angular decoding:

$$\underbrace{(z_{p_{2i}}, z_{p_{2i+1}})}_{\substack{\text{latent variables} \\ S_1 \subset \mathbb{R}^2}} \quad \mapsto \quad \underbrace{\alpha_i = \operatorname{atan2}(z_{p_{2i+1}}, z_{p_{2i}}) + \pi}_{\substack{\text{angle} \\ [0, 2\pi]}} \quad \mapsto \quad \underbrace{p_i = 2\frac{\alpha_i}{2\pi} - 1}_{\substack{\text{position} \\ [-1, 1]}} \tag{12}$$

where $S_1 \subset \mathbb{R}^2$ is the unit circle in the plane and $\operatorname{atan2}(y, x)$ returns the angle of the vector $(x, y)$ in the range $[-\pi, \pi]$. For the mapping of the cognitive variables $\mathbf{c}$ to their respective latents $\mathbf{z_c}$, we use exactly the same procedure. Generally, we consider $P$ positional variables and $C$ cognitive variables. For position, we have $\mathbf{p} \in [-1, 1]^P$, leading to $\mathbf{z_p} \in S_1 \times S_1 \times \ldots \times S_1 \subset \mathbb{R}^{2P}$. With the addition of cognitive variables, we have $\mathbf{c} \in [-1, 1]^C$, leading to $\mathbf{z} = (\mathbf{z_p}, \mathbf{z_c}) \in S_1 \times S_1 \times \ldots \times S_1 \subset \mathbb{R}^{2P+2C}$. For many of the visualizations we consider a one-dimensional position and contextual variables $p, c \in [-1, 1]$ and therefore $\mathbf{z} = (\mathbf{z_p}, \mathbf{z_c}) \in S_1 \times S_1 \subset \mathbb{R}^4$, a 4-d torus.

**Additional details and justification:** We note that classic models of place cells can be interpreted as having a periodic representation of space [130]. We stress that in the more general case of a multi-dimensional latent variable representation, periodic codes do not necessarily mean that neural activity will be periodic—this will only come about in the special case that all latent variables have either the same period, or periods that divide evenly into one another. Indeed, this is precisely why periodic grid cell input can account for localized, non-periodic place cell activity [63,84], as we also see in the grid realignment ED remapping model (Fig 3g–3l), though the approximate periodicity induced can account for place cells with multiple fields in large environments [27,28]. The addition of non-periodic cognitive variables will further diversify activity (Fig 4). This formulation could be made more general by not strictly constraining trajectories to a hypersphere or torus, but also allowing for magnitude changes reflecting gain modulation [132]. We note that while for simplicity we modeled cognitive variables with an angular code as well, there may be reasons to model them with other geometries (e.g., linear [104]).

## 2. Examples of remapping

Following the general formulation defined in Sect 1, we now provide additional details about the three types of remapping described in Eq 8. We keep these descriptions at an abstract level here, in an attempt to emphasize the independence of these descriptions from the particular RNN architecture that we employ for the examples (whose details we discuss in the following Section).

**2.1. Encoder-decoder (ED) remapping.** ED remapping features changes in $\mathbf{E}^A$ for each environment $A$. In these cases, for simplicity, we assume there are no cognitive variables being

encoded, i.e., $C = 0$ and therefore $Z = P$ and $\mathbf{z} = \mathbf{z_p}$. This results in the stable map for environment $A$

$$\mathbf{r}^A(\mathbf{p}) = \mathbf{E}^A \mathbf{z_p}(\mathbf{p}) + \boldsymbol{\nu}(\mathbf{z_p}(\mathbf{p}); \mathbf{E}^A). \tag{13}$$

To constrain how the $\mathbf{E}^A$ matrices change across environments, we define an *embedding space* that contains all subspaces reached by $\mathbf{E}^A$ for all environments $A$. Given variable $\mathbf{y}$ in the embedding space, we can then write the mappings between neural and latent spaces as

$$\underbrace{\mathbf{z}_p(\mathbf{p})}_{\substack{\text{latent space} \\ \mathbb{R}^Z}} \overset{\text{env. specific}}{\longleftrightarrow} \underbrace{\mathbf{y}(\mathbf{p})}_{\substack{\text{embed. space} \\ \mathbb{R}^Y}} \overset{\text{fixed}}{\longleftrightarrow} \underbrace{\mathbf{r}(\mathbf{p})}_{\substack{\text{neural space} \\ \mathbb{R}^N}}, \tag{14}$$

with dimensionalities $Z \leq Y \leq N$. Similar to the latent variables, the embedding variables can be linearly decoded from neural activity via the decoder matrix $\mathbf{D_y} \in \mathbb{R}^{Y \times N}$. Importantly, this mapping is *fixed* across environments. It then specifies an analogous pseudo-linear encoder as

$$\mathbf{r}^A(\mathbf{p}) = \mathbf{E_y} \mathbf{y}^A(\mathbf{p}) + \boldsymbol{\nu}_\mathbf{y}(\mathbf{y}^A(\mathbf{p})), \tag{15}$$

with $\mathbf{E_y} = \mathbf{D_y^+} \in \mathbb{R}^{N \times Y}$. The remapping across environments is then constrained to the mapping between latent and embedding spaces, $\mathbf{z}_p \leftrightarrow \mathbf{y}$. We define this as an *environment-specific* linear mapping

$$\mathbf{y}^A(\mathbf{p}) = \mathbf{R}^A \mathbf{z_p}(\mathbf{p}). \tag{16}$$

We note that the $\mathbf{z} \leftrightarrow \mathbf{y}$ mapping is linear in both directions—$\mathbf{R}^A \in \mathbb{R}^{Y \times Z}$ expands the dimensionality and then can be inverted via its left pseudo-inverse. Overall, this then allows us to decompose the parameters of the pseudo-linear encoder equation from Eq 13 as

$$\mathbf{E}^A = \mathbf{E_y} \mathbf{R}^A, \tag{17}$$

and a null space function

$$\boldsymbol{\nu}(\mathbf{z_p}(\mathbf{p})) = \boldsymbol{\nu}_\mathbf{y}(\mathbf{R}^A \mathbf{z_p}(\mathbf{p})). \tag{18}$$

In sum, the full mapping $\mathbf{E}^A \in \mathbb{R}^{N \times Z}$ is decomposed first into the variable mapping from $\mathbf{z}$ to $\mathbf{y}$ via $\mathbf{R}^A \in \mathbb{R}^{Y \times Z}$ followed by the fixed mapping from $\mathbf{y}$ to $\mathbf{r}$ via $\mathbf{E_y} \in \mathbb{R}^{N \times Y}$. The embedding dimensionality $Y$ thus constrains the relative dimensionalities of the fixed versus environment-specific components of the mappings. Typically, we choose $\mathbf{E_y}$ to be random (see Sect 3), but the choice of $\mathbf{R}^A$ depends on the two specific implementations of ED remapping, which we now describe.

### 2.1.1. Multi-chart ED remapping: random $\mathbf{R}$

We first adapted the above embedding space formulation to the case of random mappings across environments analogous to the multi-chart model [15]. We sample mappings randomly as

$$\mathbf{R}^A \sim U_{ortho}(Y \times Z), \tag{19}$$

where $Y$ is a hyperparameter that sets the *dimensionality* of the embedding space, as explained above, and $U_{ortho}(Y \times Z)$ is the uniform distribution of matrices $Y \times Z$ orthonormal in the columns. We consider two cases for $Y$:

**Low-dimensional case, $Y<N$ (Fig 3a–3f):** By constraining $\mathbf{y}$ to be low-dimensional, we arrive at a generalization of the multi-chart model where environments share a common embedding space that limits the randomness of the remapping.

**High-dimensional case, $Y = N$ (S4 Fig):** Like in the original model [15], we can consider the case in which the association between each neuron and its preferred position is made as random as possible across environments. There are two additional technical comments about this model and the relationship with the multi-chart model. First, in order to enforce full linear independence between all neurons, we not only set $Y = N$, but also enforce $\mathbf{E_y}$ to be fully orthonormal, and as such, for simplicity, we choose $\mathbf{E_y} = \mathbf{Id}_N$ (i.e., the $N \times N$ identity matrix), and therefore $\mathbf{E}^A = \mathbf{R}^A$. Second, unlike the original model, neurons are not arranged to be on a grid with uniform place field size. In principle this could be implemented by further constraining $\mathbf{R}^A$ to have "equally spaced" rows, but we do not impose this here.

### 2.1.2. Grid realignment ED remapping: module phase shift $\mathbf{R}$

Following related models from the literature [63,84], we then adapted the ED remapping formulation to the case of grid realignment. Instead of randomly mapping a low-dimensional latent trajectory $\mathbf{z}$ into a higher-dimensional embedding space $\mathbf{y}$, grid realignment works by expanding the dimensionality of $\mathbf{z}$ itself, and then modulating trajectories in that space (i.e., $Y = Z$). Thus, instead of using a single angular module to encode $\mathbf{p} \in \mathbb{R}^P$ in $\mathbf{z_p} \in \mathbb{R}^{2P}$ (Eqs 11 & 12), we use $m>1$ modules, such that $\mathbf{z_p} \in \mathbb{R}^{2mP}$. This leads to a modified angular encoding with $m$ modules as (S7 Fig)

$$\underbrace{p_i}_{\substack{\text{position} \\ [-1,1]}} \mapsto \underbrace{(\alpha_{i,j})_{j=1}^{m}, \alpha_{i,j} = \left(\frac{3}{2}\right)^{f_j} \pi(p_i + 1)}_{\substack{m \text{ angles} \\ [0,(\frac{3}{2})^{f_1} \cdot 2\pi] \times \ldots \times [0,(\frac{3}{2})^{f_m} \cdot 2\pi]}} \mapsto \underbrace{\mathbf{z_p} = (\cos(\alpha_{i,1}), \sin(\alpha_{i,1}), \ldots, \cos(\alpha_{i,m}), \sin(\alpha_{i,m}))}_{\substack{2m \text{ latent variables} \\ S_1 \times \ldots \times S_1 \subset \mathbb{R}^{2m}}}$$

(20)

where $f_j$ is the frequency of the $m$th module. For simplicity, we enforce $f_1 = 0$, which means the first module will be restricted to $\alpha_1 \in [0, 2\pi]$ and form a bijection $\mathbf{p} \in [-1,1] \longleftrightarrow \mathbf{z} \in S_1$. This simplification allows us to use the first module as a proxy for spatial position itself. In turn, the angular decoding simplifies to (S7 Fig)

$$\underbrace{(z_{p,i,1}, \ldots, z_{p,i,2m})}_{\substack{2m \text{ latent variables} \\ S_1 \times \ldots \times S_1 \subset \mathbb{R}^{2m}}} \mapsto \underbrace{\alpha_{i,1} = \text{atan2}(z_{p,i,1}, z_{p,i,2}) + \pi}_{\substack{\text{angle} \\ [0, 2\pi]}} \mapsto \underbrace{p_i = 2\frac{\alpha_{i,1}}{2\pi} - 1}_{\substack{\text{position} \\ [-1,1]}}$$

(21)

We note that the grid module scheme as we use it here induces a square grid lattice. We made this choice for simplicity, but we note that our model can be straightforwardly extended to a "twisted" torus topology [133], which would correspond to the hexagonal geometry observed in grid cell data [91].

From the perspective of the embedding space formulation above, we can now again consider remapping through an appropriate choice of the linear mapping matrix $\mathbf{R}^A$ for each environment $A$. Specifically, we can model grid realignment in our $m$ grid modules as sampling $m$ phase shift matrices $(\mathbf{R}_j^A)_{j=1}^m$ and compose them together as

$$\mathbf{R}^A = \begin{pmatrix} \mathbf{R}_1^A & \mathbf{0}_{2\times 2} & \cdots & \mathbf{0} \\ \mathbf{0}_{2\times 2} & \mathbf{R}_2^A & \cdots & \mathbf{0} \\ \vdots & \vdots & \ddots & \vdots \\ \mathbf{0} & \mathbf{0} & \cdots & \mathbf{R}_m^A \end{pmatrix},$$

(22)

where

$$\mathbf{R}_j^A \sim Rot(2 \times 2) = Rot(\Delta\alpha_{i,j}) = \begin{pmatrix} \cos(\Delta\alpha_{i,j}) & -\sin(\Delta\alpha_{i,j}) \\ \sin(\Delta\alpha_{i,j}) & \cos(\Delta\alpha_{i,j}) \end{pmatrix} \qquad (23)$$

where $Rot(2 \times 2)$ is the uniform distribution over rotational matrices $2 \times 2$ (S7b Fig).

**Interpretation of grid realignment and decoding:** We consider a simplified decoding scheme for grid realignment where the first grid module is restricted to a single period within $[-1,1]$. This allows us to decode position solely from the first module (Eq 20). From this perspective, the additional grid modules become analogous to the cognitive variables of MS remapping, causing activity changes due to the mixed selectivity of each place cell for multiple modules. We note that this choice of decoding does not affect the resultant representations, and our framework and results are also consistent with more sophisticated decoding methods from multiple grid modules [52].

**2.2. Mixed-selective remapping.** MS remapping features changes in $\mathbf{z}^A(\mathbf{p})$ for each environment $A$, leading to the stable map

$$\mathbf{r}^A = \mathbf{E}\mathbf{z}^A + \boldsymbol{\nu}(\mathbf{z}^A) \qquad (24)$$

for each environment $A$. We considered two cases of MS remapping: (i) space-feature coding, with both positional and environmental variables $P \neq 0 \ \wedge \ C \neq 0$, and (ii) implicit-space coding, with only environmental variables $P = 0 \wedge C \neq 0$ (akin to purely sensory/memory models of remapping).

*2.2.1. Space-feature coding: $P \neq 0$*

For space-feature coding, we assume that position is coded by a spatial latent trajectory $\mathbf{z_p}$, which is shared across environments, along with cognitive latent variables $\mathbf{z_c}^A$, which are specific to each environment $A$. This means we can rewrite Eq 24 as

$$\mathbf{r}^A = \mathbf{E_p}\mathbf{z_p} + \mathbf{E_c}\mathbf{z_c}^A + \boldsymbol{\nu}((\mathbf{z_p}, \mathbf{z_c}^A)). \qquad (25)$$

Perhaps the most crucial component of MS remapping is how the cognitive environmental variables and their corresponding latent trajectories are generated. Since we do not model a particular task or state-space model of an environment here, the best we can do is to sample trajectories from a random process similar to other published models [64,86]. Specifically, we define $C$ cognitive variables as

$$\mathbf{c}^A(\mathbf{p}) = \mathbf{k}^A + g^A(\mathbf{p}) \qquad (26)$$

with vector $\mathbf{k}^A \in [-1, 1]^C$ and gaussian process $g^A(\mathbf{p}) \sim GP(\mathbf{0}, K) \in [-1, 1]^C$ with kernel $K$. With this formulation we can use $\mathbf{k}^A$ to model constant or mean changes in variables across environments, and $g^A(\mathbf{p})$ to model smooth, position-dependent fluctuations. In practice, we defined a single variance parameter $\sigma$ that affects both mean and variance of the cognitive variables. Specifically, we use

$$k^A \sim \begin{cases} \mathcal{N}(0, \sigma) & \text{if } \sigma < 1 \\ U([-1, 1]) & \text{if } \sigma = 1 \end{cases} \qquad (27)$$

for each component $k^A$ of $\mathbf{k}^A$, and we use a GP with kernel

$$K(\mathbf{x}, \mathbf{x}') = \sigma v^2 e^{-\frac{\|\mathbf{x}-\mathbf{x}'\|^2}{2v^2}}. \qquad (28)$$

To make sure the cognitive variables stay bounded, $c_i \in [-1, 1]$, we constrain them consistently within a $S_1$ latent space, i.e.: $c_i \leftarrow (c_i + 1) \mod 2 - 1$.

In the final section of the results (Fig 6), we use space-feature coding to simulate a reward location task. For this case, we designed the cognitive variables to reflect the presence or absence of a reward. Specifically, we used only one cognitive variable, $c(p)$, and had it follow a Gaussian whose mean, $\mu$, reflected the location of the reward in this particular environment, and whose standard deviation, $\sigma$, reflected the spread of the reward,

$$c(p) = e^{-\frac{(p-\mu)^2}{2\sigma^2}} - 1. \tag{29}$$

*2.2.2. Implicit-space coding: P = 0*

For implicit-space coding, we instead consider that there is no explicit representation of space, i.e., $P = 0$, and therefore "place" cells only arise from the diverse, position-dependent variation in the non-spatial, cognitive variables [33,35]. In this case we can rewrite Eq 24 as

$$\mathbf{r}^A = \mathbf{E_c}\mathbf{z}_\mathbf{c}^A + \boldsymbol{\nu}(\mathbf{z}_\mathbf{c}^A). \tag{30}$$

We generate the cognitive variables in the same way as in the space-feature case.

**2.3. Null-space remapping.** Third, we consider cases that fall into the category of null-space (NS) remapping, i.e., changes in the network lead to changes in $\boldsymbol{\nu}^A$ that lead to changes in $\mathbf{r}^A$. In this final case the stable map can be written as

$$\mathbf{r}^A = \mathbf{E}\mathbf{z} + \boldsymbol{\nu}^A(\mathbf{z}). \tag{31}$$

As described in the main text, NS remapping is by definition outside of the latent space formulation, and thus we can only specify more details about how it works if we specify a concrete network model, which will impose structure within and outside of the latent space. Specifically, in Sect 3.2.2, we consider a case where we modify the non-linear component $\boldsymbol{\nu}^A(\mathbf{z})$ by modulating neural thresholds or excitability.

## 3. Simulations

To simulate a concrete model of the remapping cases described above, we used a recurrent neural network (RNN) model based on the spike-coding network (SCN), an efficient spiking autoencoder network with competitive interactions between neurons [57,59]. To retain the most generality, in the following we consider that the input and output coding of the network is in terms of the embedding space $\mathbf{y}$, rather than the latent space $\mathbf{z}$. For most of the models, these are equivalent as $\mathbf{y} = \mathbf{z}$, except for the multi-chart ED remapping model. In this section, we first discuss the input given to the network, then the RNN model and its parameters, and finally additional aspects of decoding and visualizations.

**3.1. Network input (encoding).** For the simulations, we discretized spatial position $\mathbf{p}$ into an equally-spaced grid, and then computed the cognitive variables $\mathbf{c}(\mathbf{p})$ on this grid, using the methods explained above. Both positional and cognitive variables were then transformed into the latent variables $\mathbf{z}^A = (\mathbf{z_p}, \mathbf{z}_\mathbf{c}^A)$ using the angular encoding. In order to put all three remapping types into the same framework, we finally transformed the latent variables into the embedding space, using

$$\mathbf{y}^A = \mathbf{R}^A\mathbf{z}^A. \tag{32}$$

For encoder-decoder remapping, we used the definitions of $\mathbf{R}^A$ defined above. For mixed-selective remapping, we simply used $\mathbf{R}^A = \mathbf{Id}_Y$, thereby making $\mathbf{y}^A = \mathbf{z}^A$. Lastly, we simulated null-space remapping using a single environment from the multi-chart ED remapping model, and thus we followed the formulation for ED remapping.

Finally, the embedding variable underwent an additional normalization step to ensure accurate autoencoding in the network model [59]. In cases of MS remapping with space-feature coding (Sect 2.2.1) where $\mathbf{y} = \mathbf{z} = (\mathbf{z_p}, \mathbf{z_c})$, we normalized each part separately such that $\|\mathbf{z_p}\| = \|\mathbf{z_c}\| = 1/\sqrt{2}$. When the dimensionality of $\mathbf{y} \in \mathbb{R}^Y$ is large, we additionally rescaled the embedding variables according to $\mathbf{y} \leftarrow \sqrt{Y}\mathbf{y}$.

### 3.2. Recurrent neural network (RNN) model.

#### 3.2.1. Network dynamics

The spike-coding network (SCN) was originally formulated as a spiking neural network that encodes a set of time-varying signals $\mathbf{y}(t)$ such that they can be linearly decoded from exponentially-filtered spikes of the network $\mathbf{r}(t)$ via the linear readout $\mathbf{y} = \mathbf{D_y}\mathbf{r}$. The optimal network architecture contains a low-rank recurrent weight matrix defined as $-\mathbf{D_y}^\top\mathbf{D_y}$ and input weights $\mathbf{D_y}^\top$, which constrain dynamics to a low-dimensional space [134]. The spiking neurons also have thresholds defined as $\mathbf{T}$. The optimal choice is $\mathbf{T} = \frac{1}{2}\text{diag}(\mathbf{D_y}^\top\mathbf{D_y})$. The steady-state firing rates of the SCN model can be rephrased as a convex optimization problem [58,87,135]. Instead of simulating the full spiking network, we can approximate the steady-state firing-rate dynamics by solving for the optimization problem. Specifically, for a set of discrete positions $\mathbf{p}_i$ (see Sect 3.2.1 below) we use CVXPY [136] to solve the constrained optimization problem:

$$\mathbf{r}^*(\mathbf{p}_i) = \arg\min_{\mathbf{r}} \|\mathbf{y}(\mathbf{p}_i) - \mathbf{D_y}\mathbf{r}\|^2 + 2\mathbf{T}^\top\mathbf{r}$$
$$\text{subj. to} \quad \mathbf{r} \geq \mathbf{0} \tag{33}$$

using $\mathbf{y}(\mathbf{p}_i)$ as a parameter and obtaining the feasible optimal solution $\mathbf{r}^*(\mathbf{p}_i)$. This is convenient in that it allows for more efficient simulations of the network, as our aim in this work was to characterize the steady-state rate maps across different environments. We note that in the future this framework could be easily extended to model realistic time-varying trajectories either using this rate-based formulation or the original spiking formulation.

#### 3.2.2. Network Parameters: decoding weights

The primary network parameters are the decoding weights $\mathbf{D_y} \in \mathbb{R}^{Y \times N}$. These parameters were chosen randomly as

$$\mathbf{D_y} \sim U_{norm}(Y \times N), \tag{34}$$

where $U_{norm}$ represents the uniform distribution over the set of matrices that have normalized columns. The only exception to this was the special case of full-dimensional multi-chart remapping ($Y = N$), where we instead set $\mathbf{E_y} = \mathbf{D_y} = \mathbf{Id}_N$ (see Sect 2.1.2). Following this, we applied additional normalization to the decoding weights on a case by case basis. To do so, we use the notation $\mathbf{D}_i$ for $i \in [1, ..., N]$ to refer to the $i$th column of $\mathbf{D}$, i.e., neuron $i$'s decoding weights.

We then differentiated two general normalization schemes applied to the random decoding weights. A mixed code (M) is one that simply follows the column normalization of Eq 34 above, i.e.,

$$\|\mathbf{D}_i\| = 1, \quad \forall i \in 1 ... N. \tag{35}$$

A conjunctive code (C) is one in which each pair of latent variables (corresponding to individual angles or environmental variables) is normalized following

$$||\mathbf{D}_{i,2j:2j+1}|| = 1/\sqrt{Y/2}, \quad \forall i \in [1,..,N], \forall j \in [0,...,Y/2-1], \tag{36}$$

where $\mathbf{D}_{i,2j:2j+1}$ refers to a consecutive pair of rows (i.e., embedding dimensions) in column $i$. We note that Eq 36 also imposes that Eq 35 is true, thereby making the conjunctive code a more constrained case of a mixed code. A conjunctive code imposes that each neuron's tuning vector has equal magnitude for both position and cognitive variables, leading to circular tuning in angle space (Fig 1g, inset; Fig 6b, left). In contrast, a mixed code instead implies a random direction in 4-d latent space, which could mean tuning to one of the two angles, resembling the more ellipsoid tuning as schematized in Fig 6b, right. We can also interpret these two codes using the following geometrical intuition: the mixed code constrains all neural tuning vectors to lie on the $(Y-1)$-sphere, whereas the conjunctive code further constrains tuning vectors to lie on the $(Y-2)$-torus, which is a subset of the $(Y-1)$-sphere. For the case of ED remapping, we utilized an M-code for the multi-chart model as remapping can rotate trajectories on the $(Y-1)$-sphere. Null-space remapping was simulated using the multi-chart ED model, and so also used this normalization. For grid realignment, we instead used a C-code because trajectories are naturally constrained to the $(Y-2)$-torus. This also ensured that each neuron receives equal-magnitude input from all grid modules, thereby preventing stereotypical periodic firing.

For space-feature MS remapping with both positional and cognitive latent variables, we considered additional variations on these two codes by normalizing the positional and cognitive variables separately. In this case, we have $Y = Z = 2P + 2C$. We use $\mathbf{D}_i^P$ to denote the $i$th column of $\mathbf{D}$, but only the first $2P$ rows corresponding to the positional environmental variables, and $\mathbf{D}_i^C$ denoting the last $2C$ rows corresponding to the cognitive environmental variables. With this separation, we could then choose to give one set of variables a mixed code and the other set a conjunctive code, by applying Eq 36 selectively only to one variable type. For most space-feature MS simulations, we specifically used a conjunctive-mixed (CM) code in which positional variables were made conjunctive, but the cognitive variables were left as mixed. This reflects the fact that cognitive variables are less constrained. An additional step that we took to ensure that all neurons were tuned to all environmental variables, regardless of the number of cognitive variables, was to normalize the overall weights of the two variable types as

$$||\mathbf{D}_i^P|| = ||\mathbf{D}_i^C|| = 1/\sqrt{2}. \tag{37}$$

This ensured that as cognitive dimensionality was increased, neurons retained a finite spatial tuning preference.

Finally, for the reward coding example in Fig 6 we included a last, hand designed case called the Pure and Mixed (pM) code. We chose the first $pure_P$ neurons to be pure positional, i.e. $\mathbf{D}_i^C = \mathbf{0} \; \forall i \in [1,...,pure_P]$, the next $pure_C$ neurons to be pure cognitive, i.e., $\mathbf{D}_i^P = \mathbf{0} \; \forall i \in [pure_P,...,pure_P + pure_C]$ and the rest were drawn as a mixed code defined above. This ad-hoc choice was done in order to "simulate" the effect of having a large network with random high-dimensional tuning (in which some neurons would end up appearing pure-selective by chance; Fig 6, right) while keeping the network size small and comparable to the conjunctive case (Fig 6, left). The issue with simulating a more random mixed code directly is that, in practice, the competitive winner-take-all nature of the RNN architecture we employed ensures that purely-tuned neurons are almost always out-competed by other neurons. Thus,

our ad-hoc pM code also serves as a hypothetical example of what a more feedforward architecture with less recurrent competition would produce.

### 3.2.3. Thresholds: null-space remapping in the SCN model

We note that the set of neural thresholds **T** appear in the objective function in the second linear regularization term, which induces sparsity [135]. While these parameters are typically considered to be fixed and equal across neurons, they can be varied to control the excitability of individual neurons, resulting in changes to the resultant solution that are all compatible with accurate coding [59,67]. This is precisely the definition of null-space remapping that we introduce here, and thus we can use these threshold parameters as a direct control over these type of variations. We also note that such threshold changes are mathematically equivalent to input currents applied to each neuron's voltage in the spiking version of the model [59,67]. This allows a more direct comparison between NS and MS remapping—MS remapping involves changes in input currents along low-dimensional coding directions, whereas NS remapping involves possibly unconstrained changes in inputs that lie outside of these coding dimensions.

As mentioned above, thresholds **T** are typically set to fixed and equal values across neurons. In NS remapping, however, these parameters were modified to induce rate changes. Specifically, based on a sparsity parameter $spar \in [0,1]$ we choose $N \cdot spar$ neurons, and set their thresholds $\mathbf{T}_{suppr} = 10$ and repeat the simulation (approximately 10x their normal values to prevent spiking). We note that the "default" map features one half of the neurons at elevated thresholds (making them silent) and we model "cell birth" as a release of this inhibition, allowing them to participate in the map.

**3.3. Network output (decoding).** Once we obtain a firing-rate trajectory $\mathbf{r}^A$, we decode an estimate $\hat{\mathbf{y}}^A = \mathbf{D}\mathbf{r}^A$ followed by the latent variables $\hat{\mathbf{z}}^A = \mathbf{R}^T\hat{\mathbf{y}}^A$. We then use angular decoding (Eq 12) to arrive at decoded estimates of the environmental variables $(\mathbf{p}, \mathbf{c})$. We note that for implicit-space coding, we cannot obtain estimates of position directly without training an additional decoder, as there was no explicit position encoded into the network.

**3.4. Figures and simulations parameters.** We include parameters for all simulations and plots in supplementary S1 and S2 and S3 Tables.

**3.5. Visualizations.** To visualize rate fields in angle space (e.g. Fig 3i) we consider a $200 \times 200$ mesh grid of points in the square $[-1,1]^2$, and then encode them on a torus as explained in Methods Sect 1.3. We then color code the original mesh grid with a color determined by the most excited neuron, i.e., $\arg\max_j \tilde{\mathbf{r}}^{(i)}_{j=1,...,N}$. For the suppression setup in the null-space remapping we set the suppressed neurons rates to zero before computing the argmax, i.e. $\mathbf{r}^{(i)}_{j\in Supp} \leftarrow \mathbf{0}$. Finally, for the multi-chart case, since we do not use a torus, but a sphere, we use the gnomonic projection to get the $\mathbf{z}^{(i)}$ to visualize the sphere: $\mathbf{z} = (2x, 2y, x^2 + y^2 - 1)/(x^2 + y^2 + 1)$.

Since solving the full optimization problem for each position separately was computationally expensive, we approximated the firing rates (for the illustrative cartoons only) by the feedforward input to each neuron, $\tilde{\mathbf{r}}^{(i)} \approx \mathbf{D}_\mathbf{y}^\top \mathbf{y}^{(i)}$.

**3.6. Code availability.** The code for all simulations and analysis presented here can be found in the following GitHub repository: https://github.com/guillemarsan/RemappingGeometry

## 4. Data analysis

Remapping simulations yielded a set of $N$-dimensional firing-rate maps $\mathbf{r}^A(\mathbf{p})$ for each position $\mathbf{p}$ within each environment. We then used these rate maps to compute the overlap and

spatial correlation measures used to assess the randomness of each model. As a preprocessing step, we first thresholded all rate maps to set small values less than $10^{-3}$ to zero. We computed mean rates for each environment by averaging over all positions, denoted $\bar{\mathbf{r}}^A$.

We used cosine similarity to compute the overlap between environments. Using the definition $S_c(\mathbf{x}, \mathbf{y}) = \mathbf{x} \cdot \mathbf{y}/\|\mathbf{x}\|\|\mathbf{y}\|$ for the cosine similarity, the total overlap between two environments $A$ and $B$ was given as

$$\omega(A, B) = S_C\left(\bar{\mathbf{r}}^A, \bar{\mathbf{r}}^B\right). \tag{38}$$

We then computed a shuffled overlap as

$$\omega_{\text{shuff}}(A, B) = \left\langle S_C\left(\bar{\mathbf{r}}^A_{\text{shuff}}, \bar{\mathbf{r}}^B_{\text{shuff}}\right)\right\rangle_{\text{shuff}}, \tag{39}$$

where $\bar{\mathbf{r}}^A_{\text{shuff}}$ indicates a version of $\bar{\mathbf{r}}^A$ where the elements (i.e., neuron identities) have been randomly shuffled, and $\langle \cdot \rangle_{\text{shuff}}$ indicates an average over shuffle realizations (computed over 20 realizations for each pair of environments). Given a set of environments, $\{E_1, \ldots, E_K\}$, we then obtained means of these two measures over all pairs of environments as

$$\bar{\omega} = \left\langle \omega(E_i, E_j)\right\rangle_{i,j} \tag{40}$$

$$\bar{\omega}_{\text{shuff}} = \left\langle \omega_{\text{shuff}}(E_i, E_j)\right\rangle_{i,j}, \tag{41}$$

where $\langle \cdot \rangle_{i,j}$ indicates an average over pairs of environments for which $i \neq j$.

We refer to an individual neuron's rate map in environment $A$ over all simulated positions as $\mathbf{r}^A_i$, where the elements of the vector now contain the neuron's firing rate at the different positions. We then computed the average rate map spatial correlation between two environments as

$$\rho(A, B) = \left\langle S_C\left(\mathbf{r}^A_i, \mathbf{r}^B_i\right)\right\rangle_i, \tag{42}$$

where $\langle \cdot \rangle_i$ indicates an average over all neurons that were active in both environments. We then computed an analogous shuffle control as

$$\rho_{\text{shuff}}(A, B) = \left\langle S_C\left(\mathbf{r}^A_i, \mathbf{r}^B_j\right)\right\rangle_{ij}, \tag{43}$$

where now the average was over randomly chosen pairs of neurons $i$ and $j$, again ensuring that both neurons were active in both environments. Finally, given a set of environments, $\{E_1, \ldots, E_K\}$, we again obtained means of these two measures over all pairs of environments as

$$\bar{\rho} = \left\langle \rho(E_i, E_j)\right\rangle_{i,j} \tag{44}$$

$$\bar{\rho}_{\text{shuff}} = \left\langle \rho_{\text{shuff}}(E_i, E_j)\right\rangle_{i,j}, \tag{45}$$

where now $\langle \cdot \rangle_{i,j}$ indicates an average over pairs of environments for which $i \neq j$.

To check if the overlap and spatial correlation were different from the shuffle control, we followed approaches from the literature [18,19] and used a one-sample t-test to compare the distributions of means against the overall shuffle mean. Then we concluded significance (marked with '*' in the Figures) if the p-value $< p_{thresh}$. For the parameter sweeps we use a Bonferroni correction where we set $p_{thresh} = 0.05/n$ where $n$ is the number of tests (i.e., number of individual parameter settings in the sweep).

## 5. Summary table

**5.1. Encoder-decoder (ED) remapping.** Most studies observing activity changes that are qualitatively described as "complete" remapping are most likely indicative of ED remapping. Several experimental studies have observed seemingly random remapping in area CA3 consistent with full-D or high-redundancy ED remapping [12,18–20], and that representations become more orthogonalized with experience [8,9,137]. We note that most of these studies only considered a pair of environments (with the exception of [20]), which our simulations suggest are insufficient to identify non-random population structure (S3 Fig). There is some evidence that CA1 yields less random remapping more consistent with low-redundancy or low-dimensional population geometry [12,18]. Other studies have observed progressive changes between two distinct maps, also consistent with low-dimensional population structure [138,139]. Another recent study introduces the concept of "re-registration" of common population structure to multiple environments [6,47], which we also interpret as evidence of low-dimensional ED remapping.

The classic multi-chart attractor is the definitive model of random remapping [15,16]. Other models have proposed grid realignment-like mechanisms [32,63,84]—our results suggest that these models will generate structured non-random remapping (Fig 3l), consistent with preserved place cell-grid cell relationships predicted from one of these models [32]. In another line of work, some studies have proposed a latent inference framework where multiple maps can be optimally combined [39,140], in line with the progressive morphing experiments mentioned above [138]. Though we do not consider it here, such a framework could be compatible with ED remapping with a shared low-dimensional embedding space that combines multiple maps. Lastly, another recent computational study [93] has proposed mechanisms for context-invariant sensory representations that resemble the rotational modulations of ED remapping.

**5.2. Mixed-selective (MS) remapping.** Studies observing partial remapping are typically difficult to reconcile with ED remapping and are much more compatible with MS remapping. This includes early remapping studies reporting partial and rate remapping under some conditions [8,12]. In addition, many studies have demonstrated selectivity of place cells for non-spatial information (e.g., [21,23–26,46,61,77–79,90]), which is consistent with MS remapping.

Computational models that explicitly incorporate non-spatial coding via conjunctive position-sensory representations [32,86], prospective state information [30], or representations without explicit positional information [33,38,64] can be interpreted as space-feature and implicit-space MS remapping, respectively. However, many of these models feature recurrent computations or looped interactions between different representations that pose conceptual difficulties with our representational framework [32,35–37]. As mentioned above, such latent inference frameworks argue that remapping is never due directly to sensory changes, but rather an updated inference about the cognitive map more akin to ED remapping [39]. This highlights an important limitation of the representational framework presented here, which places all of these important computational questions into the latent trajectories fed as input to the network. Thus while our framework can be made consistent with all of these models, it requires precise knowledge about the representational space they employ.

**5.3. Null-space (NS) remapping.** Short-timescale (e.g., trial-to-trial) variability in place cell activity [99] and longer-timescale drift in hippocampal populations [81,82] are compatible with NS remapping. Some studies have explicitly proposed a role for excitability [80,83], which supports our mechanism of participation modulation. However, other work suggests a component of these changes may involve synaptic plasticity [119], which we would place under the category of ED remapping. Another study demonstrated the appearance of a new,

non-overlapping map upon full suppression of a place field map [68]. Our work proposes a novel explanation of this experiment in terms of NS remapping (Fig 5h), though we note that behavioral changes were reported that may suggest a change of map more akin to ED remapping.

Some computational models have proposed excitability or other input changes consistent with NS remapping, but these are often also accompanied by plasticity [88,119]. The spike-coding network framework utilized in this work has long proposed mechanistic explanations for trial-to-trial variability, cell death, and experimental inhibition [57,59,67,87]. A recent study has explicitly made the connection between this architecture and slow changes in excitability as an account for representational drift in sensory systems [89]. We would classify all of these SCN studies as purely NS remapping. Outside of the hippocampal literature, experimental and computational studies of working memory and motor control have long found a role for null space activity [62,65,101], providing evidence that our framework could be extended to other brain areas and computations.

## Supporting information

**S1 Fig. Place cell statistics for a single environment. a,b**: Mean place field size (as % of the environment; **a**) and percentage of active place cells in each environment (**b**) as a function of redundancy ($N/Y$) for three different dimensionalities ($Y$) for multi-chart ED (mean across neurons and 10 environments, with SEM across environments). Example place fields shown in (**a**) plotted for ($Y$, redundancy) = $(64, 1)$ and $(64,64)$. Related to the distribution of percentage of neurons active in $n$ rooms (see S3c Fig).
(TIFF)

**S2 Fig. Comparison of remapping changes in each component of pseudo-linear encoder.** Norm of the remapping vector between 10 environments computed as total ($||\mathbf{r}^A - \mathbf{r}^B||$); spatial ($||\mathbf{D_p}\mathbf{r}^A - \mathbf{D_p}\mathbf{r}^B||$, which is equal to $||\mathbf{y}^A - \mathbf{y}^B|| = ||(\mathbf{R}^A - \mathbf{R}^B)\mathbf{z_p}||$ for ED and NS remapping and $||\mathbf{z_p}^A - \mathbf{z_p}^B||$ for MS remapping); cognitive ($||\mathbf{D_c}\mathbf{r}^A - \mathbf{D_c}\mathbf{r}^B|| = ||\mathbf{z_c}^A - \mathbf{z_c}^B||$); and NS ($||(\mathbf{r}^A - \mathbf{r}^B) - \mathbf{E}((\mathbf{z}^A - \mathbf{z}^B)|| = ||\boldsymbol{\nu}^A - \boldsymbol{\nu}^B||$). Shown for main figure examples of multi-chart ED remapping (**a**), grid realignment ED remapping (**b**), space-feature MS remapping (**c**), implicit-space MS remapping (**d**), and NS remapping (**e**).
(TIFF)

**S3 Fig. Multi-chart encoder-decoder (ED) remapping analysis. a**: Overlap (solid) and shuffle overlap (dashed) between 10 (left) and 30 (right) environments as a function of redundancy ($N/Y$) for different dimensionality ($Y$) values. Stars mark where the mean overlap is significantly different from the shuffle mean (t-test, Bonferroni correction with $n$ = 3 for full-D and $n$ = 21 for low-D, see Methods Sect 4). The full-dimensional embedding case ($N/Y$ = 1; Methods Sect 2.1.1; S4 Fig) is plotted at an x-axis value of -1 to differentiate it from the other cases (blue, orange, green). Note the difference significance levels for left versus right by changing the amount of data (10 versus 30 environments). **b**: Spatial correlation, same as in (a). Note that none of the data is significantly different from random in this case. **c**: Histogram of percentage of neurons active in $n$ rooms for different redundancies, measured over 10 rooms as in left panels in (**a**, **b**).
(TIFF)

**S4 Fig. Multi-chart encoder-decoder (ED) remapping with full-dimensional (full-D) embedding space. a**: Example place field rate maps for two environments. **b**: Overlap and spatial correlation distributions for 10 rooms, with mean (black) and comparison with a

shuffle control (red), showing consistency with truly random remapping. Related to full-D simulations (plotted at x-axis value of -1) from S3a Fig and S3b Fig.
(TIFF)

**S5 Fig. Space-feature mixed-selective (MS) remapping analysis. a**: Overlap (solid) and shuffle overlap (dashed) between 10 (left) and 30 (right) environments as a function of redundancy ($N/Y$) for different dimensionality ($Y$) values. Stars mark where the mean overlap is significantly different from the shuffle mean (t-test, Bonferroni correction with $n = 20$, see Methods Sect 4). **b**: Overlap (solid) and shuffle overlap (dashed) between 10 (left) and 30 (right) environments as a function of dimensionality ($Y$) for different values of cognitive variables variance ($\sigma$, see Methods Sect 2.2.1). Stars mark where the mean overlap is significantly different from the shuffle mean (t-test, Bonferroni correction with $n = 20$, see Methods Sect 4). Note the difference significance levels for left versus right in panels (**a**, **b**) by changing the amount of data (10 versus 30 environments). **c**: Spatial correlation, same as in (a). **d**: Spatial correlation same as in (b).
(TIFF)

**S6 Fig. Null-space remapping analysis. a, b**: Overlap and spatial correlation (solid) along with shuffle controls (dashed), comparing "default" network (50% sparsity) to NS remapping where other amounts of sparsity are chosen (*spar*, see Methods Sect 3.2.2); mean and SEM computed for 5 random selections of suppressed neurons in each case. Stars mark where the mean overlap is significantly different from the shuffle mean (t-test, Bonferroni correction with $n = 6$, see Methods Sect 4). **c**: Example trajectories and place fields visualized in angle space (top) and as a function of position (bottom) for different levels of sparsity, following panels (**a**, **b**). Three neurons highlighted (1, 2, & 3) highlighting dropping in and out, and small tuning modulations. For all panels, note that *spar* > 50% indicates cell birth and *spar* < 50% indicates suppression or cell death.
(TIFF)

**S7 Fig. Grid realignment: encoding and decoding with multiple phase-shifted modules. a**: Exemplified encoding from a single environmental variable $p_i \in [-1, 1]$ to its latent representation $(z_{p_{2mi}}, z_{p_{2m(i+1)-1}}) \in S_1 \times \ldots \times S_1$ through angular encoding and corresponding decoding. Here using $m = 3$ modules with frequency parameters $f_1 = 0$, $f_2 = 1$ and $f_3 = -1$. Decoding is done only using the first module due to the restriction $f_1 = 0$. **b**: Exemplified embedding from this latent representation $(z_{p_{2mi}}, z_{p_{2m(i+1)-1}}) \in S_1 \times \ldots \times S_1$ to the embedding space $\mathbf{y} \in S_1 \times \ldots \times S_1$ through multiplication with a phase shift matrix $\mathbf{y} = \mathbf{R}\mathbf{z_p}$ (see Eq 22), constructed with smaller rotation matrices $\mathbf{R}_1, \mathbf{R}_2, \mathbf{R}_3$.
(TIFF)

**S1 Table. Simulation parameters for encoder-decoder (ED) remapping.** The notation $2^{[a,b]}$ stands for all powers of $2^j$ with integers $j \in [a, b]$.
(PDF)

**S2 Table. Simulation parameters for mixed-selective (MS) remapping.** The notation $2^{[a,b]}$ stands for all powers of $2^j$ with integers $j \in [a, b]$.
(PDF)

**S3 Table. Simulation parameters for null-space (NS) remapping.** The notation $2^{[a,b]}$ stands for all powers of $2^j$ with integers $j \in [a, b]$.
(PDF)

## Acknowledgments

We thank Hanne Stensola and Daniel McNamee for helpful discussions about our model. We also thank members of the Machens Lab for constructive feedback and comments at various stages of this work.

## Author contributions

**Conceptualization:** Guillermo Martín-Sánchez, Christian K. Machens, William F. Podlaski.

**Formal analysis:** Guillermo Martín-Sánchez, William F. Podlaski.

**Funding acquisition:** Christian K. Machens.

**Investigation:** Guillermo Martín-Sánchez, William F. Podlaski.

**Methodology:** Guillermo Martín-Sánchez, Christian K. Machens, William F. Podlaski.

**Project administration:** Christian K. Machens.

**Software:** Guillermo Martín-Sánchez.

**Supervision:** Christian K. Machens, William F. Podlaski.

**Writing – original draft:** Guillermo Martín-Sánchez, William F. Podlaski.

**Writing – review & editing:** Guillermo Martín-Sánchez, Christian K. Machens, William F. Podlaski.

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
