## [Decision Letter · Decision Letter 0]

11 Jun 2025

PCOMPBIOL-D-25-00382

Three types of remapping with linear decoders: a population-geometric perspective

PLOS Computational Biology

Dear Dr. Podlaski,

Thank you for submitting your manuscript to PLOS Computational Biology. After careful consideration, we feel that it has merit but does not fully meet PLOS Computational Biology's publication criteria as it currently stands. Therefore, we invite you to submit a revised version of the manuscript that addresses the points raised during the review process. The reviewers make good recommendations for how to clarify the model and the paper more generally.

Please submit your revised manuscript within 60 days Aug 11 2025 11:59PM. If you will need more time than this to complete your revisions, please reply to this message or contact the journal office at ploscompbiol@plos.org. Please include the following items when submitting your revised manuscript:

We look forward to receiving your revised manuscript.

Kind regards,

Daniel Bush

Academic Editor

PLOS Computational Biology

Joseph Ayers

Section Editor

PLOS Computational Biology

**Additional Editor Comments:**

In particular, the authors should endeavour to more thoroughly describe the observable features predicted by each potential mechanism for remapping, so that they might be distinguished in experimental data, and to discuss how modelling latent position in non-periodic / two-dimensional environments would affect these results.

**Journal Requirements:**

3) We notice that your supplementary Figures are included in the manuscript file. Please remove them and upload them with the file type 'Supporting Information'. Please ensure that each Supporting Information file has a legend listed in the manuscript after the references list.

Potential Copyright Issues:

i) Figures 1B, 1D, 1G, 1H, 2D, 3G, 4A, 4G, 5D, and 6A. Please confirm whether you drew the images / clip-art within the figure panels by hand. If you did not draw the images, please provide (a) a link to the source of the images or icons and their license / terms of use; or (b) written permission from the copyright holder to publish the images or icons under our CC BY 4.0 license. Alternatively, you may replace the images with open source alternatives. See these open source resources you may use to replace images / clip-art:

**Reviewers' comments:**

Reviewer's Responses to Questions

**Comments to the Authors:**

Reviewer #1: This manuscript introduces a novel theoretical framework to understand hippocampal remapping through a population geometry and neural coding perspective. The authors propose three distinct mechanisms of remapping based on linearly-decodable latent space representations: encoder-decoder (ED) remapping, mixed-selective (MS) remapping, and nullspace (NS) remapping. The framework offers a unifying approach to interpret the diverse experimental observations and theoretical models in the field.

The work is significant as it attempts to bridge different perspectives on hippocampal function and provides a systematic way to categorize various remapping phenomena. The mathematical formalism is valuable and could potentially be applied beyond the hippocampus to understand neural variability more broadly.

My only major concern is that the methods section is poorly structured and was hard to read, please:

- Add conceptual overview subsections before mathematical details

- Create a glossary of symbols

- Add schematic diagrams of the mathematical constructs

- Refer to specific subsections of the methods in the results section so you do not have to read through everything to find what you are looking for.

- Although I liked the first part of the results, including Fig 1, consider including more of the methods before the main results, particularly how the results were obtained (optimization, cost, etc)

Other than this, the manuscript could be published as is, but I include some additional concerns that I believe would increase the quality and can be taken as suggestions rather than requests.

Experimental validation and predictions:

The manuscript lacks direct validation against experimental data. While the authors discuss how their framework relates to existing experimental findings, they do not test their model predictions against specific datasets. This limits the immediate impact of the work. The necessity of angular coding for generating place-like fields could be expanded.

Suggestion: Incorporate analyses of published datasets that demonstrate how the proposed remapping mechanisms explain observed neural activity patterns. For example, how does the distribution of field sizes compare to experimental data? How does the correlation structure between environments change as a function of noise amplitude? Compare the predictions of encoder-decoder remapping with changes in environment geometry such as in Krupic et al 2015, which also reminds me that the grid cells used for grid alignment seems square, it would be nice to plot these grid cells to show their tuning.

Nullspace remapping:

The nullspace remapping mechanism is the most novel contribution but could get more attention. The authors do not fully explore the implications of this mechanism, particularly:

- How does the nullspace component depend on the choice of encoder/decoder pairs?

- What transformations of the nullspace function ν(·) would leave the decoding invariant?

- What constraints on the nullspace would be most biologically plausible?

Suggestion: Expand the analysis of nullspace remapping, perhaps including a case where ν is restricted to be linear, which would build intuition while maintaining analytical tractability. Investigate how nullspace remapping would manifest if E were trained through learning rather than fixed.

The trial-to-trial variability described in the nullspace remapping mechanism seems to create a potential inconsistency with experimental observations.

If nullspace remapping were the only mechanism at play, would it not suggest that animals would show remapping on every trial, regardless of familiarity or novelty? If so, this contradicts well-established experimental findings where Place cells show stable spatial firing fields across multiple trials in familiar environments, global remapping is typically observed when environments change significantly, and stability of place cell representations increases with familiarity. This would be worth discussing.

Reviewer #2: In this work, the authors provide a geometrical interpretation of remapping, a widely observed phenomenon where place cells change their firing properties (such as firing rate or preferred firing location) in response to contextual changes. They propose a population-level view of position coding as a trajectory in a low-dimensional latent space plus an off-manifold, non-linear component, and then use this model to define three possible mechanisms that could underlie remapping at the population level.

The effort to synthesize existing models of remapping (such as attractor maps or latent space shifts) into a unified, population-centric framework is appreciated and relevant to the hippocampal community. The paper is clearly written, and the figures are well-structured. It is evident that the authors made an effort to guide the reader through their framework using clear illustrations. However, in its current form, the paper neither provides a comprehensive theoretical account nor yields clear insights into remapping mechanisms observed in real data. Below are a few comments that I recommend addressing before the manuscript can be considered for publication.

* If the authors aim to present a theoretical framework, then a more thorough exploration of the phenomenology is needed. Specifically, which observable features serve as clear fingerprints of each remapping type, and which are merely consequences of modeling choices (such as the number of neurons, RNN implementation, angular latent space, etc.)? For example, the first (encoder-decoder) and second (mixed-selectivity) remapping types appear to produce different outcomes in terms of spatial overlap and correlations, but this is illustrated only with a single underwhelming simulation instead of investigated on a systematic level. The framework could be leveraged to extract heuristics that distinguish remapping types based on population-level properties (e.g. tuning overlap and correlation), offering testable predictions for future work and experimental validation.

* Relatedly, the lack of any application to real data limits the scope of the paper. While a computational paper can stand on its own - and I believe this paper could very well do so if the previous point is addressed - the framework would be significantly strengthened by showing how these remapping fingerprints appear (or fail to appear) in experimental datasets. There is a wealth of open-access hippocampal recordings that could be used for this purpose.

* I also find the beginning of the Results section unnecessarily dense. The initial decoding-based formulation feels superfluous, especially since the encoding model r=Ez+v already contains all the necessary components to introduce and distinguish the three remapping types. The encoding view is not only more intuitive from a neuroscience perspective but also more directly tied to how remapping is typically conceptualized. Starting from encoding would make the paper much easier to follow and more logically structured.

* The choice to model latent position as an angular (periodic) variable is also worth revisiting. This might be reasonable for certain experimental settings (such as circular mazes or virtual environments with repeating structure), but it seems problematic to assume that all spatial coding relies on periodic representations. Is this assumption required to produce place fields or to differentiate remapping types? If so, the implications should be discussed. If not, it would be helpful to show that the results hold in a non-periodic latent space as well.

* Finally, I am not sure what is gained from the set of single-example simulations (e.g., cell silencing, reward modulation, etc.). These feel disconnected from the core framework and do not offer clear conceptual insight. Rather than testing a large number of scenarios each loosely mapped to a different experiment, the paper would benefit more from a focused and systematic analysis of the population-level consequences of each remapping strategy (see my first point). Such an analysis would help ground the framework in observable phenomena and make it more useful to the community.

Overall, the paper presents an interesting perspective and an appealing geometric formulation. However, I believe that with a more systematic exploration of the core model, a clearer presentation, and at least a minimal link to data, the paper could make a much stronger and more lasting contribution.

**Have the authors made all data and (if applicable) computational code underlying the findings in their manuscript fully available?**

Reviewer #1: None

Reviewer #2: Yes

PLOS authors have the option to publish the peer review history of their article (what does this mean?). If published, this will include your full peer review and any attached files.

Reviewer #1: No

Reviewer #2: No

**Figure resubmission:**
---

## [Decision Letter · Decision Letter 1]

22 Sep 2025

Dear Dr. Podlaski,

We are pleased to inform you that your manuscript 'Three types of remapping with linear decoders: a population-geometric perspective' has been provisionally accepted for publication in PLOS Computational Biology.

Best regards,

Daniel Bush

Academic Editor

PLOS Computational Biology

Marieke van Vugt

Section Editor

PLOS Computational Biology

Reviewer #1:

Reviewer #2:

Reviewer's Responses to Questions

**Comments to the Authors:**

Reviewer #1: My thanks to the authors for their detailed and thoughtful revisions, and for comprehensively answering all of my questions. The manuscript is much improved, and I am happy to recommend it for publication in its current form.

Reviewer #2: The authors clearly took the reviews seriously and added a significant amount of work and detail to the paper. The manuscript is now more complete, clearer, and provides a much more thorough exploration of the different modeling choices and parameters. I particularily appreciate the addition of systematic parameter sweeps and the inclusion of a survey of experimental data and how they interface with the proposed set of models. The revised version addresses my main concerns and I recommend acceptance of the revised version.

**Have the authors made all data and (if applicable) computational code underlying the findings in their manuscript fully available?**

Reviewer #1: Yes

Reviewer #2: None

PLOS authors have the option to publish the peer review history of their article (what does this mean?). If published, this will include your full peer review and any attached files.

Reviewer #1: No

Reviewer #2: No

---

## [Editor Report · Acceptance letter]

PCOMPBIOL-D-25-00382R1

Three types of remapping with linear decoders: a population-geometric perspective

Dear Dr Podlaski,

I am pleased to inform you that your manuscript has been formally accepted for publication in PLOS Computational Biology. Your manuscript is now with our production department and you will be notified of the publication date in due course.

With kind regards,

Anita Estes
